# Conserved functional antagonism of CELF and MBNL proteins controls stem cell-specific alternative splicing in planarians

Jordi Solana[1]*[†], Manuel Irimia[2,3]*[†], Salah Ayoub[1], Marta Rodriguez Orejuela[1], Vera Zywitza[1], Marvin Jens[1], Javier Tapial[2,3], Debashish Ray[4], Quaid Morris[4,5], Timothy R Hughes[4,5], Benjamin J Blencowe[4,5], Nikolaus Rajewsky[1]*

[1]Systems Biology of Gene Regulatory Elements, Berlin Institute for Medical Systems Biology, Max-Delbrück Center for Molecular Medicine, Berlin, Germany; [2]Centre for Genomic Regulation, Barcelona Institute of Science and Technology (BIST), Barcelona, Spain; [3]Universitat Pompeu Fabra, Barcelona, Spain; [4]Donnelly Centre, University of Toronto, Toronto, Canada; [5]Department of Molecular Genetics, University of Toronto, Toronto, Canada

**Abstract** In contrast to transcriptional regulation, the function of alternative splicing (AS) in stem cells is poorly understood. In mammals, MBNL proteins negatively regulate an exon program specific of embryonic stem cells; however, little is known about the *in vivo* significance of this regulation. We studied AS in a powerful *in vivo* model for stem cell biology, the planarian *Schmidtea mediterranea*. We discover a conserved AS program comprising hundreds of alternative exons, microexons and introns that is differentially regulated in planarian stem cells, and comprehensively identify its regulators. We show that functional antagonism between CELF and MBNL factors directly controls stem cell-specific AS in planarians, placing the origin of this regulatory mechanism at the base of Bilaterians. Knockdown of CELF or MBNL factors lead to abnormal regenerative capacities by affecting self-renewal and differentiation sets of genes, respectively. These results highlight the importance of AS interactions in stem cell regulation across metazoans.

*For correspondence:
jordisolana@gmail.com (JS);
mirimia@gmail.com (MI);
rajewsky@mdc-berlin.de (NR)

[†]These authors contributed equally to this work

## Introduction

Stem cells are found in all animals and are defined by their capacity to self-renew and to differentiate into different cell types (*Sánchez Alvarado and Yamanaka, 2014*). In mammals, embryonic stem cells (ESCs) derived from pre-implantation embryos can be cultured *in vitro* and differentiated into virtually any cell type (*Martello and Smith, 2014*); however, a similarly potent cell type has not been found in adults. In contrast, in other animals, pluripotent stem cells are maintained during the entire life, and are often associated with extraordinary regenerative capabilities (*Solana, 2013*; *Tanaka and Reddien, 2011*). One of the most extreme examples are freshwater planarians, from which almost any body part can regenerate a complete organism in a few days. This ability relies on a large number of stem cells present in the adult, called neoblasts. Illustrating their pluripotency, single neoblasts transplanted into lethally irradiated hosts can rescue this lethality, restore tissue turnover, generate all cell types of the adult planarian and completely transform the genotype and phenotype of the host into that of the donor (*Wagner et al., 2011*). However, recent analyses at single-cell resolution showed that the neoblast pool is highly heterogeneous, also including multiple lineage-committed precursors (*van Wolfswinkel et al., 2014*). Despite significant progress, how neoblasts are

**eLife digest** Stem cells are specialized cells found in all animals that can develop into several different types of mature cells. Stem cells are therefore well suited for maintaining organs that are in heavy use, such as the intestine, and for regenerating tissues that are prone to injury, like the skin.

One reason why stem cells differ from mature cell types is because they activate, or "express", different sets of genes. In addition, many genes can be expressed as one of several versions. These variants, also known as isoforms, are generated by a process called alternative splicing. In mature cells in mammals, a group of proteins called the MBNL proteins help to prevent the expression of gene isoforms that are characteristic to stem cells.

The adult flatworm *Schmidtea mediterranea* contains stem cells that can regenerate any part of the body. Solana, Irimia et al. have now investigated whether alternative splicing is important for controlling how the worm's stem cells behave. After establishing which gene isoforms are expressed in the stem cells and the mature cells, the levels of different sets of proteins that control alternative splicing were experimentally reduced.

The results indicate that just as seen in mammals, the MBNL proteins reduce the expression of stem cell-related gene isoforms in the flatworms. Furthermore, Solana, Irimia et al. found that another protein called CELF counteracts MBNL proteins by helping to express gene isoforms that are active in stem cells.

The interplay between the MBNL and CELF proteins has also been observed in human cells. Thus, it appears that this way of controlling alternative splicing is common to flatworms and mammals and is therefore evolutionarily ancient. This suggests that other similar ways of controlling stem cells by interactions between regulatory proteins might be working in all animal stem cells. Further studies are now needed to investigate these control proteins.

regulated and enable planarian cell turnover as well as regeneration upon wounding is still largely unknown.

Initial transcriptomic analyses of planarian neoblasts have revealed hundreds of genes that are differentially enriched in both planarian and mammalian stem cells compared to all differentiated cell types despite 500 million years of independent evolution (*Labbé et al., 2012*; *Onal et al., 2012*; *Reddien et al., 2005*; *Resch et al., 2012*; *Rouhana et al., 2010*; *Solana et al., 2012*), suggesting the existence of universal regulatory features across animal pluripotent cells. However, this conservation does not include the major transcriptional regulators of mammalian stem cells. In ESCs, pluripotency is maintained by a core set of transcription factors that include OCT4, NANOG and SOX2, but these factors and their interactions are largely not conserved beyond the vertebrate lineage (*Fernandez-Tresguerres et al., 2010*; *Gold et al., 2014*; *Onal et al., 2012*). For instance, no homolog of NANOG has been described to date in any invertebrate species, despite extensive search (*Scerbo et al., 2014*). Therefore, elucidating how the regulation of pluripotency in invertebrates occurs in the absence of this core set of factors is crucial to understand the biology of animal stem cells.

Post-transcriptional regulation is more recently emerging as another key mechanism for controlling ESC biology (*Ye and Blelloch, 2014*). In particular, various reports have established the importance of alternative splicing (AS) for ESCs and somatic cell reprogramming (*Han et al., 2013*; *Ohta et al., 2013*; *Venables et al., 2013*; *Ye and Blelloch, 2014*). AS is the process by which introns and exons are selectively included or excluded from the pre-mRNA to produce multiple mRNA and protein isoforms. AS can therefore expand transcriptomic complexity in a cell type- or developmental stage-specific manner, adding an extra layer of regulation to the control of gene expression. Moreover, highly regulated alternative exons often encode disordered regions of proteins that embed binding motifs, and thus have the potential to rewire protein-protein interactions in a context-specific manner (*Buljan et al., 2012*; *Ellis et al., 2012*).

AS is chiefly regulated by RNA binding proteins (RBPs), which are themselves often differentially expressed in a cell type-regulated manner. These factors typically bind to pre-mRNAs in a sequence- and position-specific fashion, thereby modulating inclusion or exclusion of the target alternative

sequence. For example, members of the MBNL family of RBPs are lowly expressed in mammalian ESCs, but show higher levels of expression in all profiled differentiated samples – ranging from transformed cell lines (e.g. HeLa, 293T, etc.) to highly histologically and functionally diverse adult tissue types (including multiple brain regions, muscle, liver, kidney, testis, etc.) – where they repress a program of alternative exons that are characteristic of ESCs (*Han et al., 2013*). Knockdown of MBNL in differentiated cultured cells thus induces ESC-like AS patterns and this is sufficient to enhance somatic cell reprogramming (*Han et al., 2013*). MBNL targets are involved in diverse cellular processes, from cytoskeletal dynamics to gene regulation, and include disparate actors, from protein kinases to transcriptional regulators. For instance, MBNL-regulated AS of the transcription factor FOXP1 in mouse and human embryonic stem cells changes its DNA-binding properties, so that its stem cell specific isoform (FOXP1-ES) promotes transcription of pluripotency genes and represses differentiation genes, while its canonical isoform activates genes involved in differentiation (*Gabut et al., 2011*). Consistently, FOXP1-ES is repressed by MBNL upon differentiation by direct binding to intronic regions in the pre-mRNA (*Han et al., 2013*).

Intriguingly, members of another RBP family, CELF, have been shown to regulate AS in an antagonistic manner to that of MBNL factors in mammals. For example, during heart development, CELF factors promote embryonic AS patterns, which are later replaced by adult heart AS patterns promoted by MBNL proteins (*Kalsotra et al., 2008*). Multiple exons have been shown to antagonistically respond to CELF and MBNL factors in a variety of mammalian differentiation, disease and cell culture systems (*Dasgupta and Ladd, 2012*; *Kalsotra et al., 2008*; *Lee and Cooper, 2009*; *Wang et al., 2015*). CELF factors, however, have not been linked so far to regulation of AS in stem cells.

Despite these studies, the degree to which regulation by AS is conserved or whether AS plays any role in non-mammalian stem cells is still entirely unknown. To address these questions, we here investigated the regulatory role and functional importance of AS in planarian stem cells and regeneration *in vivo*. Our results show that AS and the factors involved in its regulation, as well as their specific interactions (i.e. functional antagonism between MBNL and CELF proteins), are crucial for planarian stem cell biology and regeneration and likely a deeply conserved feature of animal stem cells.

## Results

### Genome-wide annotation of planarian intron-exon structures

We used a recent *de novo* transcriptome assembly (*Liu et al., 2013*) to annotate intron and exons in the genome of *Schmidtea mediterranea* (assembly version 3.1; *Figure 1—figure supplement 1A–B*, Materials and methods). Most planarian protein-coding genes (75%) are multiexonic (*Figure 1—figure supplement 1C*). Intron length displays a sharp bimodal distribution: whereas most introns are relatively small, with lengths centred on 57 bp, another subset ranges between 1 and 10 Kbps (*Figure 1—figure supplement 1D*). Next, we produced and compiled over 30 samples of novel and available deep coverage RNA sequencing (RNA-Seq) data from multiple sources (*Figure 1—source data 1*). These consist of FACS-isolated cell populations – including neoblast-enriched (X1), neoblast progeny-enriched (X2) and differentiated cell-enriched (Xins) fractions (*Figure 1A*) – as well as wild type and neoblast-depleted whole animals. We employed these data and previously described methodologies to comprehensively identify all types of AS in planarians and quantify their alternative sequence inclusion levels using the 'Percentage Spliced In' (PSI) metric ([*Braunschweig et al., 2014*; *Irimia et al., 2014*], methods). These approaches yielded 12,276 AS events, the majority of which (56.2%) corresponded to alternatively retained introns (*Figure 1—figure supplement 2A*). We also identified 2529 alternative exons that can be either fully included or skipped from the mRNAs, including single and multi-cassette events, as well as 262 microexons (exons of length $\leq$27 nucleotides (nt) (*Irimia et al., 2014*), 72 of which had length $\leq$15 nt) (*Figure 1—figure supplement 2A*).

### Identification of a stem cell-specific AS program in planarians

By comparing neoblasts versus differentiated cells and whole worms depleted of neoblasts (methods), we identified 246 and 256 AS events with increased and decreased alternative sequence inclusion levels in neoblasts, respectively (X1-included and X1-excluded, together referred to as

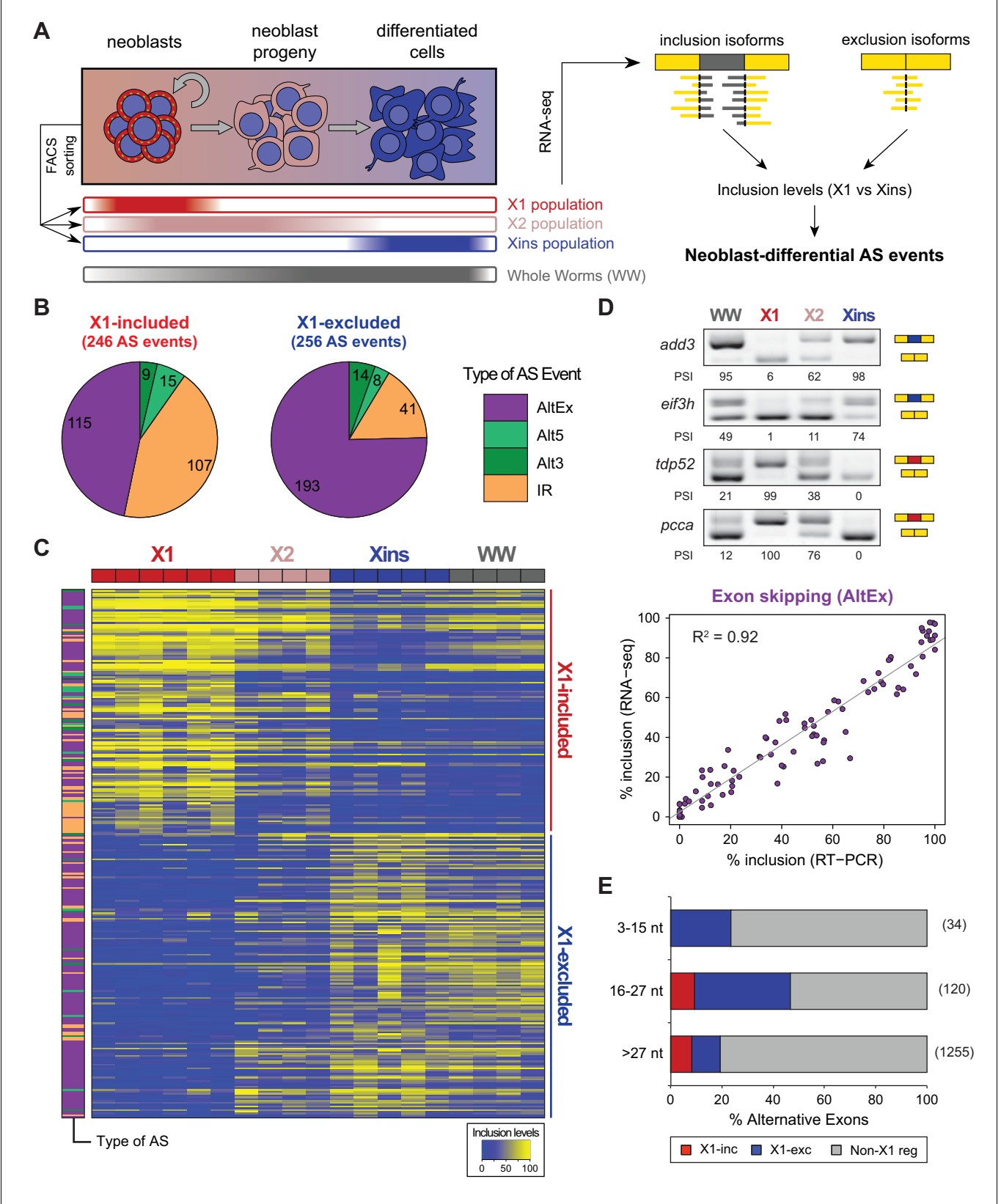

**Figure 1.** Alternative splicing is differentially regulated between planarian stem cells and differentiated cells. (**A**) Planarian Stem cells ('Neoblasts'), their differentiating progeny, and differentiated cells are purified with FACS ('X1', 'X2', 'Xins', respectively). RNA-seq and computational analyses were

*Figure 1 continued on next page*

*Figure 1 continued*

subsequently used to identify X1-differential AS at a genome-wide scale. (**B**) Distribution of AS events with increased/decreased inclusion of the alternative sequence in X1. Alt3/5, alternative splice site acceptor/donor selection; IR, intron retention; AltEx, cassette exons. (**C**) Heatmap of inclusion level values for 293 representative X1-differential AS events. Bars in the dendrogram correspond to AS types in B. (**D**) Representative RT-PCR assays monitoring AS patterns in FACS isolated cell fractions. Red and blue exons indicate those exons with higher and lower inclusion in X1 compared to Xins fractions, respectively. Scatter plot shows correspondence between PSI estimates by RNA-Seq and RT-PCR in whole worms and X1, X2 and Xins fractions for 22 events ($R^2$ = 0.92, n = 88). (**E**) Proportion of alternatively spliced exons by length class with increased inclusion in X1 ('X1-inc') or Xins ('X1-exc') fractions, or not differentially regulated between X1 and Xins fractions ('Non-X1 reg')

The following source data and figure supplements are available for figure 1:

**Source data 1.** *Schmidtea mediterranea* RNA-seq samples used in this study.
**Source data 2.** List of neoblast-differential AS events.
**Figure supplement 1.** Genome annotation pipeline and summary statistics.
**Figure supplement 2.** Identification and analysis of neoblast differential AS in planarians.
**Figure supplement 3.** RT-PCR validation of neoblast-differential AS events.
**Figure supplement 4.** Gene Ontology analysis of X1-differential AS events.

'neoblast-differential'; *Figure 1B* and *Figure 1—figure supplement 2B*, *Figure 1—source data 2*). Most of these events had intermediate inclusion levels in X2 samples (which comprise a mixture of stem cells and their early differentiation progeny, *Figure 1C*), while inclusion levels in whole worm (WW) samples largely matched those of Xins samples. This latter correlation is expected since ~80% of the cells in adult planarians are differentiated (*Baguñà and Romero, 1981*). In contrast to the global AS pattern (*Figure 1—figure supplement 2A*), the majority of neoblast-differential AS events involved cassette exons (61.4%, *Figure 1B*). RT-PCRs using primers (*Supplementary file 1*) flanking alternative exon sequences confirmed all (22/22) tested neoblast-differential exons and showed a high correlation between RT-PCR and RNA-Seq inclusion level estimates (*Figure 1D* and *Figure 1—figure supplement 3*; $R^2$ = 0.92, n = 88). Among these cassette exons, we detected 64 microexons that were differentially regulated between X1 and Xins fractions, the majority of which (82.8%) were more included in differentiated cells (*Figure 1E*). Consistent with this bias towards differentiated cells, vertebrate microexons have been recently shown to display enrichment in neural and muscle tissues compared to ESCs (*Irimia et al., 2014*).

The majority of neoblast-differential exons (68.5%) overlapped protein-coding regions and conserved the reading frame; therefore, they were predicted to generate distinct protein isoforms in neoblasts and differentiated cells. This set of AS events were significantly enriched in genes involved in cytoskeleton and cell signalling functions (*Figure 1—figure supplement 4A*), gene ontology categories that were also enriched among ESC-differential alternatively spliced genes in mammals (*Han et al., 2013*). Planarian genes with neoblast-differential exons were associated with a wide range of functions, and included cytoskeleton regulators (e.g. *add3*), membrane trafficking proteins (e.g. *tpd52*), metabolic enzymes (e.g. *pcca*), translation factors (e.g. *eif3h*) and protein kinases (e.g. *map4k3*), among others. Interestingly, neoblast-differential exons significantly more often overlapped disordered regions of proteins and avoided structured domains compared to general alternatively spliced and constitutive exons (*Figure 1—figure supplement 2C,D*; p $\leq$ 4.7 $\times$ 10$^{-5}$ for all comparisons, 3-way Fisher test and proportion tests, respectively). A similar pattern was described for tissue-specific exons in mammals (*Buljan et al., 2012*; *Ellis et al., 2012*), suggesting that neoblast-differential exons may also contribute to modulate protein-protein interactions in planarian cells.

## Unexpected abundance of neoblast-specific retained introns

Intron retention (IR) consists in the selective retention of a given intron in the mature mRNA. Notably, 107 IR events were differentially enriched in X1 fractions compared to only 41 in Xins (*Figure 1B*). RT-PCRs confirmed all (11/11) tested X1-included introns compared to half (2/4) of X1-excluded (*Figures 2A* and *Figure 1—figure supplement 3*; $R^2$ = 0.68, n = 56, for all neoblast-differential introns tested). X1-included introns were usually longer than other intron types (*Figure 2B*, P = $7.1 \times 10^{-10}$, Wilcoxon Sum Rank test), and belonged to genes that were more highly expressed in differentiated cells than in neoblasts and their progeny (*Figure 2C–D*). Importantly, in contrast to cassette exon events, which were predicted to generate different protein isoforms in neoblast and differentiated cells, nearly all X1-included IR events were predicted to disrupt the reading frame specifically in neoblasts (*Figure 2E*; p<$2.2 \times 10^{-16}$, chi-squared test). Moreover, these introns preferentially affected genes that were significantly enriched in functions related to cell differentiation, negative regulation of proliferation and cell type-specific metabolic pathways (*Figure 1—figure supplement 4B–C*). Therefore, altogether these data suggest that neoblast-specific IR may be operating to ensure that a subset of early-transcribed differentiation genes is not active in neoblasts. These genes might then be selectively activated upon differentiation by splicing out the 'detained' intron, similar to recent reports in mammalian ESCs (*Boutz et al., 2015*; *Braunschweig et al., 2014*).

## Planarian neoblast-differential AS is evolutionary conserved

To investigate whether the neoblast-differential AS program of *S. mediterranea* has been conserved during evolution, we mapped our validated set of AS exons (*Figure 1—figure supplement 3*) to a transcriptome assembly of the planarian species *Dugesia japonica* (*Nishimura et al., 2015*), which has diverged from *S. mediterranea* approximately 85 million years ago (*Lazaro et al., 2011*), similar to the estimated time for the human-mouse divergence. We used this information to design *D. japonica*-specific primers (*Supplementary file 1*), and then performed RT-PCRs in FACS isolated populations from *D. japonica* and compared them to *S. mediterranea*. Strikingly, 20/21 (95.2%) of the probed cassette exons were also present and alternatively spliced in *D. japonica*, and 16/20 (80%) of these showed conserved differential regulation between X1 and Xins fractions in the two species (*Figure 3A* and *Figure 3—source data 2*). Moreover, half of the IR events that could be probed despite the lack of a reference genome sequence in *D. japonica* also displayed conserved neoblast-differential regulation (3/6; *Figure 3A*). This level of neoblast-differential conservation is even higher than that observed between ESC-differential exons in mammals (*Han et al., 2013*), and strongly argues for the functional relevance of this AS program in planarians.

   We next asked whether neoblast-differential exons were conserved between neoblasts and human ESCs. We identified 15 orthologous gene groups with neoblast/ESC-differential exons in both planarian and humans (*Figure 3—source data 1*, Materials and methods). Remarkably, in ten of these orthologous groups, planarian and human stem cell regulated exons fall in similar regions of the protein, and in at least four of them (*add3*, *eml4*, *fmnl3* and *ppfibp1*) the intervening sequences are partly orthologous, despite extensive rearrangement of intron-exon structures (*Figure 3B* and *Figure 3—source data 2*). Therefore, although neoblast and ESC AS programs are largely lineage-specific, AS may impact some proteins in a similar manner in both species.

## CELF and MBNL factors are major regulators of planarian neoblast-differential AS

We next sought to identify *trans*-acting RBP factors that regulate neoblast-differential AS. Based on homology searches and presence of RNA binding domains, we identified over 300 putative RNA binding proteins in planarians (*Figure 4—source data 1*). Homologs of well-known mammalian tissue-specific AS factors were then selected as potential regulators of the planarian neoblast AS program (*Figure 4A*). To first assess their differential enrichment in neoblast or differentiated cell fractions, we compared their gene expression levels across our RNA-Seq panel with those of single-copy core spliceosomal components (genes associated with KEGG terms for spliceosomal complex A, B and C; *Figure 4—source data 2*). Strikingly, a member of the CELF family (*Smed-bruno-like*, *bruli*) of AS regulators showed the strongest differential expression enrichment in X1 fractions among the multiple probed AS factors (*Figure 4A* and *Figure 4—source data 1*), and had been previously reported to be needed for planarian stem cell self-renewal and regeneration (*Guo et al.,*

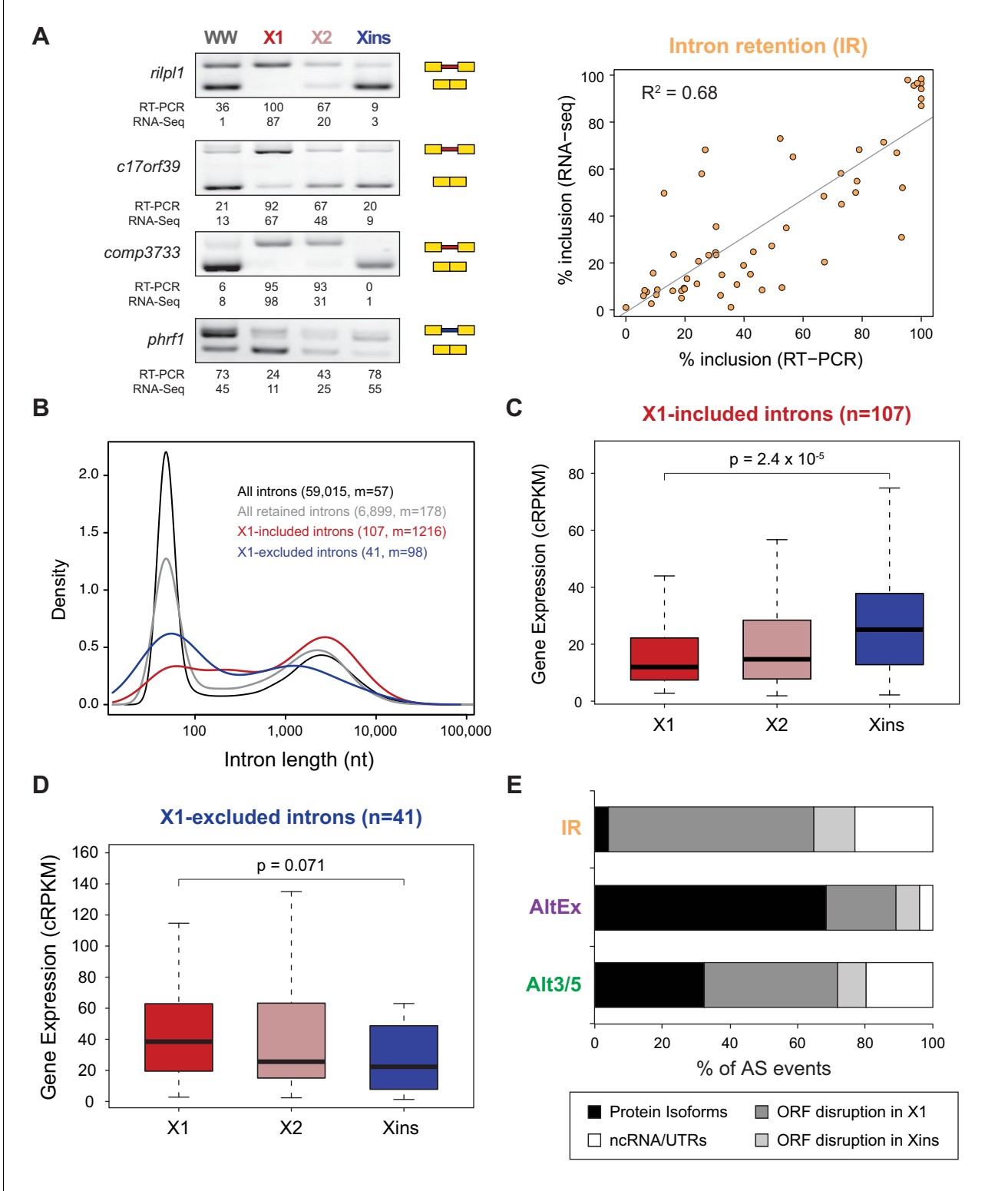

**Figure 2.** Abundant IR in planarian neoblasts. (**A**) Representative RT-PCR assays monitoring IR patterns in FACS isolated cell fractions. Red and blue introns indicate those introns with higher and lower inclusion in X1 compared to Xins fractions, respectively. Scatter plot shows correspondence between PSI estimates by RNA-Seq and RT-PCR in whole worms and X1, X2 and Xins fractions for 14 events ($R^2 = 0.68$, n = 56). (**B**) Intron length distributions for all introns (black), all retained introns (grey), X1-included introns (red) and X1-excluded introns (blue). Median length (m) is indicated for each set. (**C–D**) Gene expression measured using the cRPKM metric for genes containing X1-included (**C**) or excluded (**D**) introns in X1, X2 and Xins cell

*Figure 2 continued on next page*

*Figure 2 continued*

fractions. P-value corresponds to a Wilcoxon Sum Rank test between X1 and Xins expression values. (E) Percent of X1-differential AS events by type that are predicted to generate alternative ORF-preserving isoforms (black), disrupt the ORF in neoblasts or differentiated cells (dark/light grey), or overlap non-coding sequences (white).

*2006*). Most other investigated AS factors showed no differential enrichment in X1 fractions compared to the core spliceosome (e.g. *Smed-ptbp-1, -2, Smed-rbfox-1*), or were differentially enriched in Xins fractions (e.g. *Smed-mbnl-1, Smed-esrp-1,2*, etc.) (*Figure 4A* and *Figure 4—source data 1*).

To query the role of these factors in the regulation of the neoblast AS program and in regeneration, we performed RNAi of these genes separately or in combinations (for paralogs with similar enrichment in X1 or Xins fractions). A total of eight RNAi groups comprising two single RNAi and six multiple RNAi combinations (*Figure 4—source data 3*) were knocked down in parallel with a control

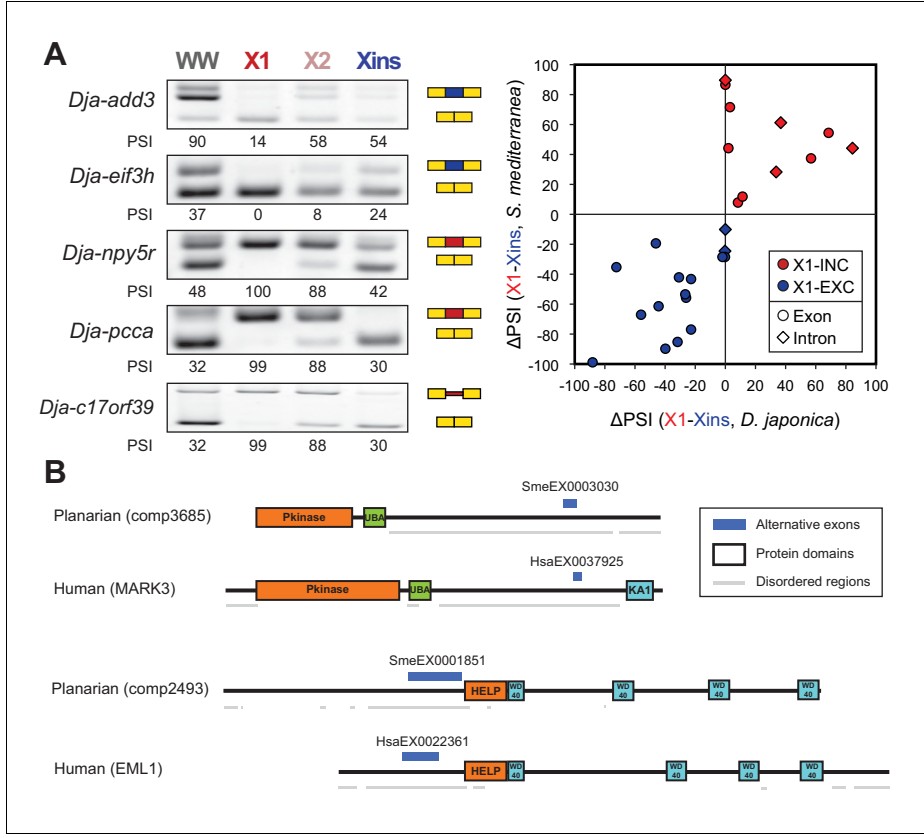

**Figure 3.** The neoblast-specific AS program is extensively conserved in *D. japonica*. (A) Left: representative RT-PCR assays monitoring AS patterns for five representative neoblast-differential AS events in FACS isolated cell fractions from *D. japonica*. Scatter plot shows correspondence between ΔPSI (X1-Xins) estimates by RT-PCR in *S. mediterranea* and *D. japonica* for 21 cassette exons (circles) and 6 retained introns (diamonds). Conservation of regulation is observed for both alternative sequences with higher ('X1-inc', red) and lower ('X1-exc', blue) inclusion in neoblasts. (B) Schematic examples of the occurrence of neoblast-differential AS (blue bars) with respect to protein domain organization in two pairs of gene homologues in human and planarian. The examples show that AS events fall in similar protein regions in both human and planarian orthologs.

The following source data is available for figure 3:

**Source data 1.** Orthologous gene groups with neoblast/ESC-differential exons in both planarian and humans.
**Source data 2.** Conservation of stem cell-differential AS events.

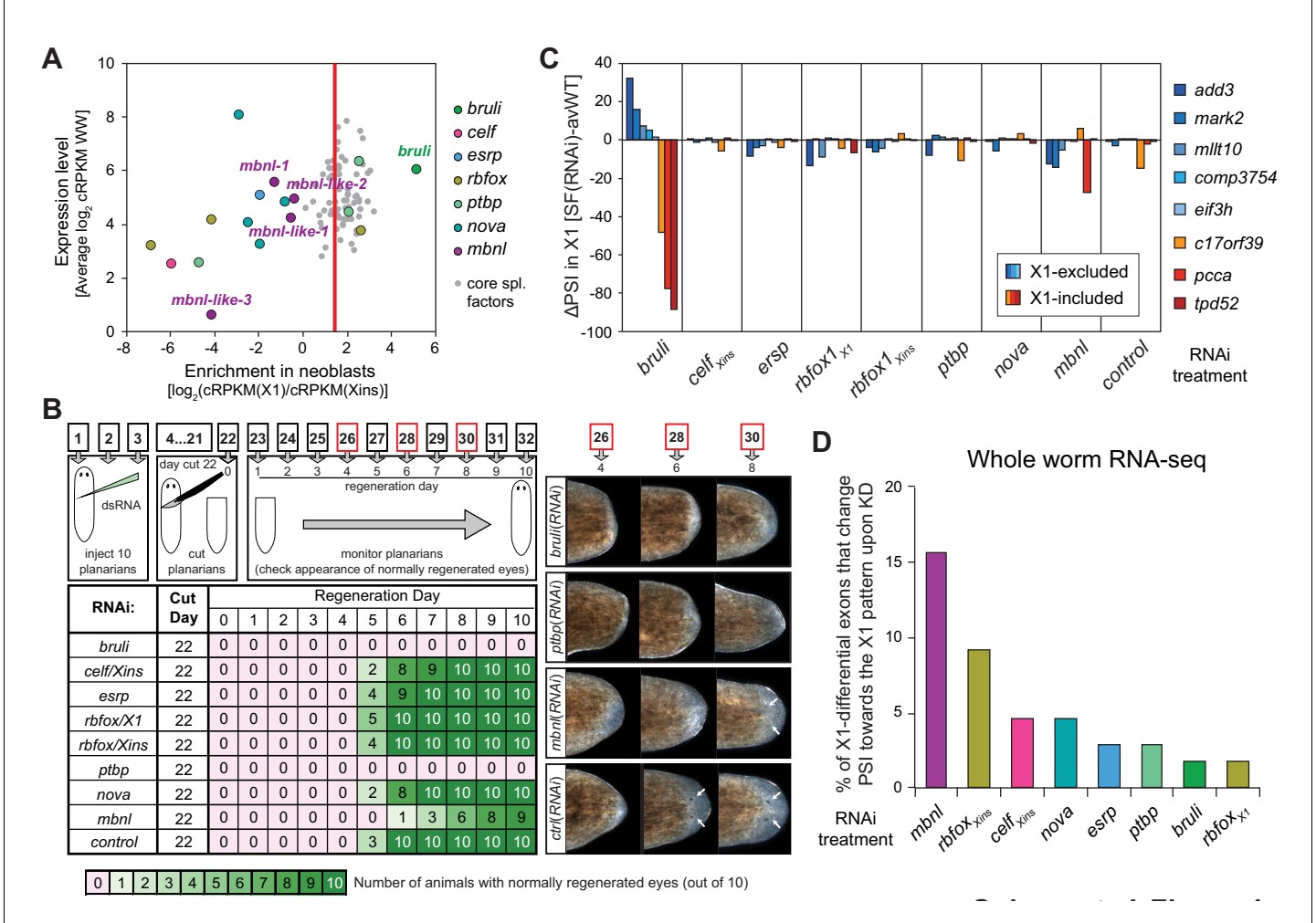

**Figure 4.** Identification of *bruli* and *mbnl* as major regulators of neoblast-specific AS. (**A**) Scattered plot highlighting differential gene expression for selected tissue-specific AS factors in planarians. X-axis: differential gene expression in X1 vs Xins cell fractions for selected AS factors (color dots) and core spliceosomal components (grey dots). The red line is the median enrichment value for the core spliceosomal components. An AS factor is thus considered to be enriched in X1 or Xins fractions if it is located to the right or to the left of this line, respectively. Y-axis: gene expression levels in whole worms (WW), using the cRPKM metric. (**B**) Test of regeneration speed after head ablation upon AS factor knockdown. Identification of normally looking eyespots was used as proxy for complete regeneration for 10 individuals per experimental condition. (**C**) ΔPSI estimates in X1 cell fractions by RT-PCR for two X1-included exons (red), one X1-included retained intron (*c17orf39*, orange) and five X1-excluded exons (blue) 10 days after RNAi treatment with dsRNA coding for AS factor combinations. ΔPSI values for each event and experiment are calculated respect to the average PSIs in three wild type samples. (**D**) Percent of neoblast-differential exons with sufficient read coverage that change their inclusion levels (ΔPSI ≥ 15) in whole worms towards the X1 pattern (as expected for a negative regulator of neoblast-differential AS) upon knockdown of each AS factor.

The following source data and figure supplements are available for figure 4:

**Source data 1.** Annotation of RBPs in planarians.

**Source data 2.** Annotation of spliceosomal components in planarians.

**Source data 3.** Single and multiple splicing factor knockdown RNAi groups.

**Figure supplement 1.** Identification of *bruli* and *mbnl* as major regulators of neoblast-specific AS.

**Figure supplement 2.** Enrichment of RBP binding motifs associated with neoblast-differential exons.

RNAi. Tests of regeneration capability upon head ablation in these knockdowns showed diverse defects for several AS factor groups (*Figure 4B*). As previously reported, *bruli*(*RNAi*) showed a dramatic phenotype with no regeneration. A similar phenotype was also observed for the *ptbp*(*RNAi*) group. Combined knockdown of the three other CELF factors, enriched in Xins fractions, induced elongation and movement defects, but did not seem to affect regeneration time dynamics. Finally, a significant delay in regeneration was observed for the *mbnl* group.

To evaluate the transcriptomic impact of these factors in regulating neoblast-differential AS in stem cells, we FACS-sorted planarian X1 cell populations from these eight RNAi groups 10 days after RNAi (when the neoblast population was still not decreased upon *bruli* RNAi treatment, *Figure 4—figure supplement 1*), and performed semi-quantitative RT-PCR for eight representative neoblast-differential AS events (*Figure 4C*). Only *bruli*(*RNAi*) treatment induced dramatic changes in inclusion levels in X1 fractions compared to the control RNAi in the expected direction (i.e. towards the differentiated pattern; *Figure 4C*), suggesting that *bruli* may act as a positive regulator of the neoblast AS program. Next, to assess the importance of the AS factors that were enriched in differentiated cells, we performed RNA-Seq of whole worms from these eight RNAi groups. Knockdown of the *mbnl* group [a four-gene knockdown: *Smed-mbnl-1;Smed-mbnl-like-1, -2, -3*(*RNAi*), hereafter *mbnl*(*RNAi*)] (*Figure 4—source data 3*) showed the most widespread changes in the X1-differential AS program towards the neoblast pattern (*Figure 4D*), suggesting that MBNL proteins may act as negative regulators of the neoblast AS program. Remarkably, RNA sequence motif enrichment analyses using a library of RNAcompete-derived binding profiles for over 100 animal RBPs (*Ray et al., 2013*) showed that MBNL consensus motifs are also the most significantly enriched in the downstream introns of X1-excluded cassette exons (*Figure 4—figure supplement 2*). This result is consistent with planarian MBNL proteins enhancing X1-excluded exons via binding to their downstream introns, as described for mammalian MBNL proteins (*Han et al., 2013*; *Wang et al., 2012*). Based on these data, we decided to investigate the role of BRULI and MBNL factors as putative positive and negative regulators of the neoblast-differential AS program, respectively.

## CELF and MBNL factors are needed for planarian regeneration

We found four potential *mbnl* orthologs in the planarian transcriptome, comprising a canonical MBNL ortholog with two pairs of zinc finger domains (*Smed-mbnl-1*, hereafter *mbnl-1*), as well as three orthologs with only one pair (*Smed-mbnl-like-1,2,3*, hereafter *mbnl-like-1,2,3*) (*Figure 5—figure supplement 1*). All *mbnl* orthologs displayed expression enrichment in Xins fractions (*Figure 4A*). Whole mount *in situ* hybridizations showed that *mbnl-1* has a widespread expression pattern, while *mbnl-like-1* and *mbnl-like-2* are expressed mainly in gut tissue (*Figure 5A*; no expression was detected for *mbnl-like-3*). Analysis of recently published single-cell sequencing data further revealed expression of *mbnl-1*, *mbnl-like-1* and *mbnl-like-2* in epidermis, which is commonly lost in whole-mount *in situ* protocols, and confirmed the widespread expression of *mbnl-1* (*Wurtzel et al., 2015*). In contrast, as previously reported (*Guo et al., 2006*), *bruli* was specifically expressed in neoblasts, as shown both by *in situ* hybridization (*Figure 5A*) and single-cell sequencing data (*Wurtzel et al., 2015*).

To further investigate the role of CELF and MBNL factors in the regulation of the neoblast AS program and regeneration, we performed RNAi of these genes, separately or in combinations (in the case of the *mbnl* paralogs). In the case of *mbnl*, a significant delay in regeneration was observed for the combined knockdown of all four *mbnl* homologues. These regeneration defects were even stronger when regenerating tails were examined (*Figure 5—figure supplement 2A*). When knocked down individually, only *Smed-mbnl-1* had minor effects on regeneration (*Figure 5—figure supplement 2A*). These observations were mirrored at the transcriptomic level, as shown by RT-PCRs for several representative AS events in whole worms (*Figure 5B*). Importantly, while the regeneration phenotype was observed 22 days after the RNAi injection, effects on AS inclusion levels where observable as soon as five days after the initiation of the treatment, plateauing after day 10 post-injection (*Figure 5—figure supplement 2B*).

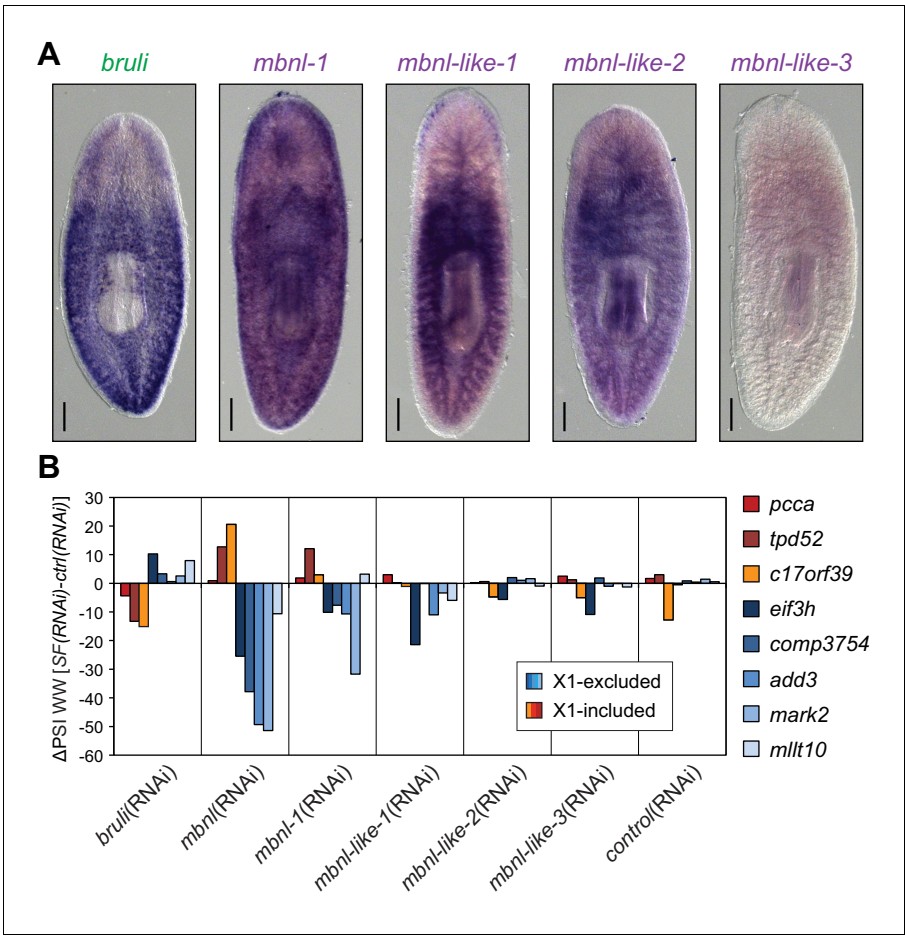

**Figure 5.** Combined *mbnl* knockdown has stronger effects than individual knockdown. (**A**) Whole worm *in situ* hybridization for *bruli*, *mbnl-1* and *mbnl-like1*, *2* and *3*. Scale bar is 0.5 mm. (**B**) ΔPSI in whole worm estimates by RT-PCR for two X1-inc exons (red), one X1-inc retained intron (c17orf39, orange) and five X1-exc exons (blue) 10 days after RNAi treatment with dsRNA coding for *bruli*, mix of dsRNAs against the four *mbnl* genes [*mbnl(RNAi)*], *mbnl-1*, *mbnl-like-1*, *mbnl-like-2*, and *mbnl-like-3*. Two independent controls samples were included; ΔPSI values are relative to the first control sample.

The following figure supplements are available for figure 5:

**Figure supplement 1.** BRULI and MBNL factors domain architecture and effects on regeneration and alternative splicing.

**Figure supplement 2.** *bruli* and *mbnl* factors effects on regeneration and alternative splicing.

## Direct antagonistic regulation of planarian stem cell-specific AS by CELF and MBNL factors

In order to investigate the concerted transcriptomic-wide impact of CELF and MBNL factors in the regulation of neoblast-differential AS, we next FACS-sorted X1 and Xins populations from *bruli* (*RNAi*), *mbnl*(*RNAi*) and *control*(*RNAi*) animals 10 days after RNAi in duplicates and subjected them to RNA-Seq. Consistent with our RT-PCR results (*Figure 4C*), *bruli* knockdown induced strong inclusion level changes in X1 fractions towards the differentiated pattern (*Figure 6A*; P = $1.4 \times 10^{-19}$, binomial test). In particular, X1-included alternative sequences became less included after *bruli* knockdown (*Figure 6A*, red dots) while X1-excluded ones were more included (*Figure 6A*, blue dots). On the other hand, *mbnl* knockdown induced changes in the opposite direction in Xins fractions (*Figure 6B*; P = $2.1 \times 10^{-11}$, binomial test). These changes in inclusion levels obtained by

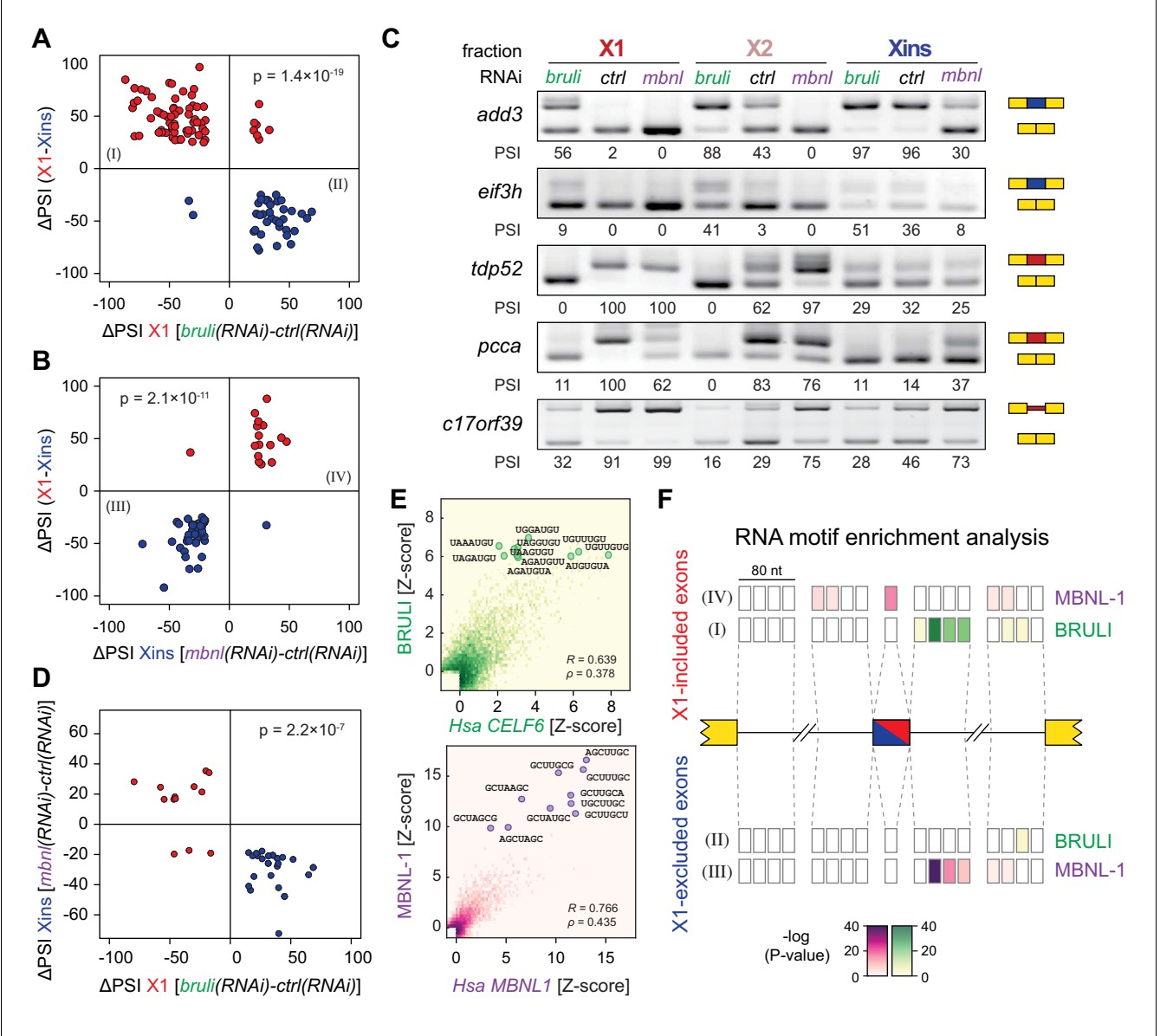

Figure 6. *bruli* and *mbnl* antagonistically regulate neoblast-specific AS. (A) High negative association (p<1.4 × $10^{-19}$, one-sided binomial test) between differences in inclusion levels (ΔPSI) of X1-differential AS events in X1 versus Xins fractions, and differences in *bruli* and control RNAi treated X1 fractions. Red/Blue dots correspond to X1-included/excluded AS events. (B) High positive association (p<2.1 × $10^{-11}$, one-sided binomial test) between differences in inclusion levels (ΔPSI) of X1-differential AS events in X1 versus Xins fractions, and differences in *mbnl* and control RNAi treated Xins fractions. Only AS events with sufficient read coverage in the control and KD samples and an absolute ΔPSI>15 are plotted. (C) RT-PCR assays monitoring AS patterns for 5 representative X1-differential AS events in FACS isolated cell fractions treated with *bruli*, control or *mbnl* RNAi. (D) Most X1-differential AS events that are affected by both *bruli* and *mbnl* knockdown are regulated in an antagonistic manner (p<2.2 × $10^{-7}$, one-sided binomial test). (E) 2-dimensional histogram of RNA-compete 7-mer Z-scores, comparing the sequence-specific binding of planarian and human proteins. Unspecific 7-mers with Z-score <0 for both RBPs were excluded. ρ, Spearman rank correlation; R, Pearson correlation. Top 10 planarian motifs are highlighted. (F) Motif-enrichment analysis. Each intronic box corresponds to a 20-nucleotide bin at the indicated location relative to the AS exon (middle box). Color encodes the significance of enrichment (Fisher's exact test, Bonferroni corrected for the number of tested bins) of high affinity 7-mers for BRULI or MBNL, comparing each differentially spliced exon set with an unaffected background set. Exon sets (I-IV) correspond to those in quadrants indicated in panels B and C.

The following figure supplements are available for figure 6:

**Figure supplement 1.** Examples of neoblast-differential AS events regulated by *bruli* and/or *mbnl*.

*Figure 6 continued*

**Figure supplement 2.** *bruli* and *mbnl* antagonistically regulate neoblast-specific AS.

RNA-seq analyses were independently validated by RT-PCR assays (19/19 changes, 100%; $R^2$ = 0.87, n = 225; *Figure 6C* and *Figure 6—figure supplement 1*). Remarkably, 33/87 (37.9%) *mbnl*-regulated AS events (14 cassette exons, 11 microexons, and 8 retained introns) were also affected by *bruli* knockdown in the opposite direction (*Figure 6D*; P = 2.2 × 10$^{-7}$, binomial test; corresponding to 33/141 (23.4%) *bruli*-regulated AS events). As expected, the number of AS changes of *bruli* and *mbnl* knockdown in Xins and X1 fractions respectively was lower and was not significantly associated with neoblast-differential regulation (*Figure 6—figure supplement 2A,B*). Of note, 37/110 (33.6%) of X1-retained introns showed decreased retention upon *bruli* knockdown (ΔPSI ≥ 15), suggesting that BRULI regulates coordinated IR events in planarian stem cells. On the other hand, *mbnl* knockdown affected a large fraction of Xins-enriched microexons (20/54, 37%).

To investigate the mechanisms by which BRULI and MBNL regulate neoblast-differential AS, we performed RNAcompete assays (*Ray et al., 2009*; *Ray et al., 2013*) to identify their consensus RNA binding motifs. Purified GST-tagged RBPs were incubated with a 75-fold excess of an RNA pool and the binding preferences of the RBP elucidated by analysing bound RNAs by microarray analyses. The top scoring motifs (*Figure 6E* and *Figure 6—figure supplement 2C*) for BRULI and MBNL-1 contained the 3-mer UGU and 6-mer GCUUGC, respectively, consistent with previous reports for their mammalian homologs. Moreover, the binding specificities of the planarian and mammalian homologs strongly correlated (*Figure 6E*) and highly significantly overlapped (*Figure 6—figure supplement 2D*; p<5.5 × 10$^{-89}$ for all pairwise homolog comparisons, hypergeometric test) across all measured 7-mers in RNAcompete assays. Thus, the planarian MBNL and CELF factors and its mammalian homologs have conserved binding specificities. Next, to investigate if planarian MBNL and CELF binding motifs were significantly associated with neoblast-differential alternative exons that show changes upon splicing factor knockdown (*Figure 6A,B*), we analysed the presence of these motifs in the regulated exons and their flanking introns compared to sets of control exons (see Materials and methods). Both BRULI and MBNL-1 motifs were highly significantly enriched in the downstream introns of exons downregulated upon the respective KD, whereas MBNL-1 motifs were also significantly enriched in exonic sequences of exons upregulated upon *mbnl* KD (*Figure 6F*). These locations are consistent with the enhancing or repressive functions, respectively, from RNA regulatory maps described for these factors in mammals (*Han et al., 2013*; *Wang et al., 2012*), and strongly suggest a direct regulatory role in AS regulation in planarians.

## CELF and MBNL factors have antagonistic effects on planarian stem cells

Next, to investigate how knockdown of *bruli* and *mbnl* impairs planarian regeneration at the cellular level, we sequenced mRNA extracted from whole worms at different time points after RNAi injections and evaluated the levels of cell type-specific markers (*Figure 7A*). It was previously described that *bruli* knockdown induces a conspicuous neoblast loss phenotype (*Guo et al., 2006*). Consistently, we observed a strong downregulation of neoblast markers after *bruli* knockdown (*Figure 7—figure supplement 1*), which were not affected by *mbnl* knockdown. Strikingly, the progeny markers *prog-1* and *prog-2-1*, which are expressed in postmitotic epidermal progenitors (*Eisenhoffer et al., 2008*; *van Wolfswinkel et al., 2014*), changed their expression levels in opposing directions upon *bruli/mbnl* knockdown. More precisely, *bruli* knockdown induced loss of these markers, whereas *mbnl* knockdown led to an increase in their expression (*Figure 7B* and *Figure 7—figure supplement 1*). Similar results were found with recently described markers for lineage-committed subclasses of neoblasts (*van Wolfswinkel et al., 2014*) (*Figure 7—figure supplement 1*) and epidermal progenitors (*Zhu et al., 2015*) (*Figure 7B*, *Figure 7—figure supplement 1*), which were confirmed by qPCR for *prog-1*, *prog-2-1* and *prog-1-1* (*Figure 7C*). This concomitant decrease and increase in the expression of multiple progeny markers for *bruli* and *mbnl* knockdown, respectively, is likely due to the loss and accumulation of progenitors, respectively, rather than to specific changes in the expression levels of these markers in an invariant pool of progenitor cells.

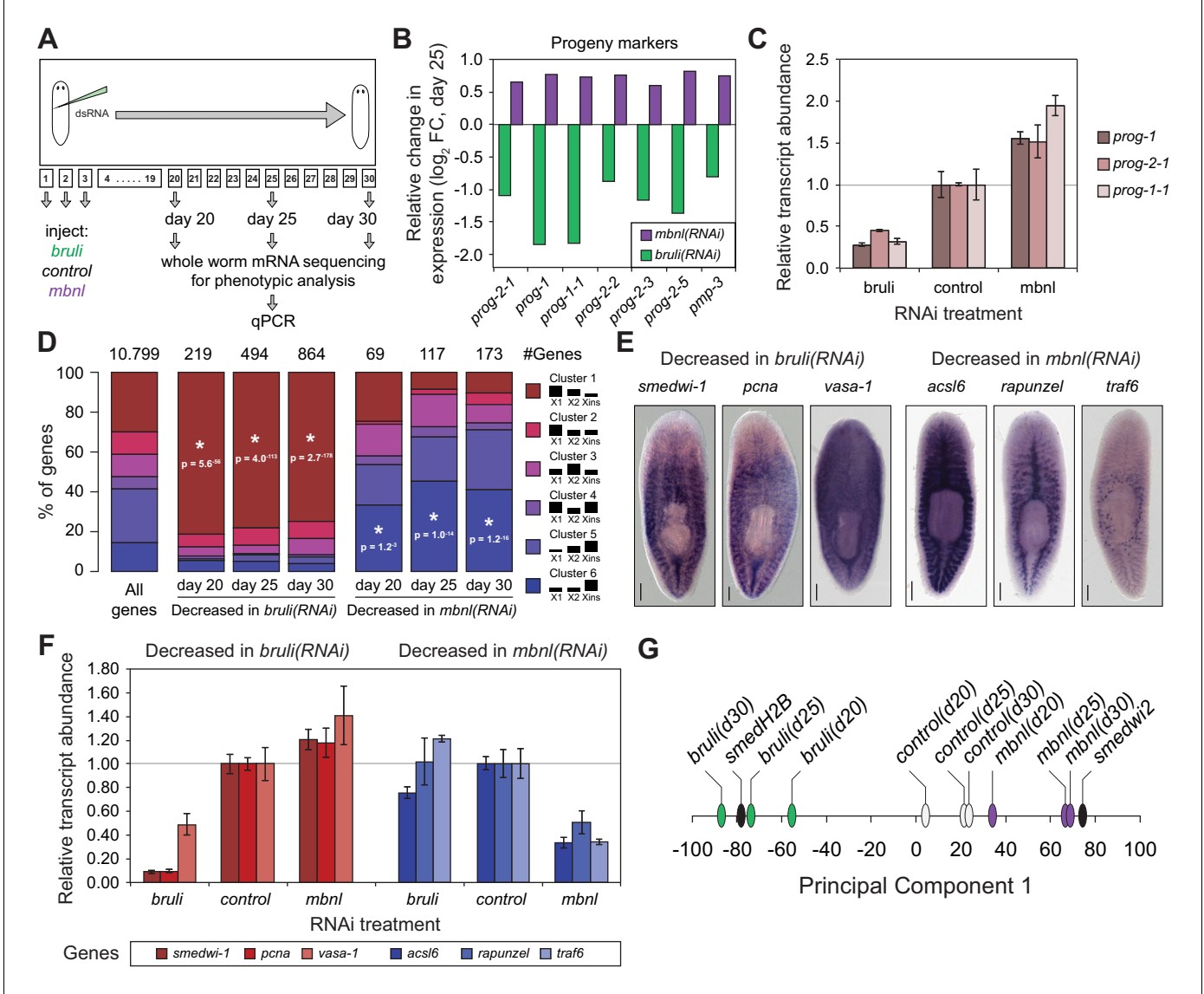

**Figure 7.** *bruli* and *mbnl* antagonistically regulate neoblast biology. (**A**) Schematic representation of phenotypic analysis experiments. (**B**) Gene expression changes estimated by RNA-Seq of several progeny markers 25 days after *bruli* (green) or *mbnl* (purple) RNAi treatment compared to controls. (**C**) Quantification of gene expression levels by qPCR of *prog-1*, *prog-2-1*, and *prog-1-1* in whole worms 25 days after treatment with *bruli*, *control* or *mbnl* RNAi. (**D**) Proportion of genes belonging to each of the 6 clusters defined in (**Onal et al., 2012**) based on their X1vs Xins enrichment that are decreased 20, 25 and 30 days after *bruli* and *mbnl* RNAi treatment. P-values correspond to hypergeometric tests for Clusters 1 (P = 5.6 × 10$^{-56}$, P = 4.0 × 10$^{-113}$, P = 2.7 × 10$^{-178}$, for *bruli*(RNAi) days 20, 25 and 30, respectively) and 6 (P = 1.2 × 10$^{-3}$, P = 1.0 × 10$^{-14}$, P = 1.2 × 10$^{-16}$, for *mbnl*(RNAi) days 20, 25 and 30, respectively). All other tests were not significant. Black histograms on the right side indicate schematically, for each cluster, the relative gene expression levels in each cell fraction. (**E**) *In situ* hybridization of three representative Cluster 1 (red bars) and Cluster 6 (blue bars) genes in whole worms. (**F**) qPCR-based gene expression estimates of three representative Cluster 1 (red bars) and Cluster 6 (blue bars) genes in whole worms 25 days after *bruli, control* or *mbnl* RNAi. (**G**) Principal Component 1 separates transcriptomes from *bruli* RNAi treated samples together with *Smed-H2B*(RNAi)(which affects neoblast self-maintenance) from *mbnl* RNAi treated samples together with *Smedwi-2*(RNAi)(which impairs neoblast differentiation).

The following figure supplements are available for figure 7:

**Figure supplement 1.** *bruli* and *mbnl* knockdown have contrasting effect of specific gene markers.

**Figure supplement 2.** Gene Ontology analysis of genes downregulated upon *bruli* and *mbnl* knockdown.

Next, we compared the effect of *bruli* and *mbnl* knockdown on the levels of gene sets previously defined by clustering common patterns of expression in X1, X2 and Xins fractions (*Onal et al., 2012*). Cluster 1 comprises genes with strong differential expression enrichment in neoblasts, while Cluster six genes are conversely expressed mainly in differentiated tissues and excluded from the neoblast compartment. Consistent with a prominent neoblast loss (*Guo et al., 2006*), Cluster 1 genes were highly significantly decreased upon *bruli* knockdown (*Figure 7D* ; $p < 5.6 \times 10^{-56}$ for the three time points, hypergeometric test). In contrast, *mbnl* knockdown impacted predominantly Cluster six genes (*Figure 7D* ; p<0.01 for the three time points, hypergeometric test). The expression patterns of three decreased genes from Clusters 1 and the from Cluster 6 were confirmed by *in situ* hybridization (*Figure 7E*). The three Cluster 1 transcripts were observed in neoblasts as previously described; on the other hand, the three Cluster six genes impacted by *mbnl* knockdown were expressed only in differentiated cells (comprising gut and secretory-like cells). We then confirmed the depletion of these markers after *bruli* and *mbnl* RNAi by qPCR. (*Figure 7F*). The three Cluster 1 transcripts were strongly decreased after *bruli* knockdown, while the three Cluster six genes were decreased only after *mbnl* knockdown. These results are thus consistent with prominent neoblast loss after *bruli* RNAi, on the one hand, and progressive scarcity of differentiated cell types due to impaired neoblast differentiation after *mbnl* knockdown, on the other hand. These opposed effects of *bruli* and *mbnl* were also reflected in the gene functions altered upon knockdown, with stem cell function-related terms enriched in *bruli*(RNAi) decreased genes, compared to differentiated tissue function associated terms for *mbnl*(RNAi) (*Figure 7—figure supplement 2*). In addition, a Principal Component Analysis (PCA) showed that the first PC sharply separated *bruli* samples together with *Smed-H2B*(RNAi)(which affects neoblast self-maintenance [*Solana et al., 2012*]) from *mbnl* samples together with *Smedwi-2*(RNAi)(which impairs neoblast differentiation [*Reddien et al., 2005*]) (*Figure 7G*), further showing their opposing effects in stem cell properties. Taken together our results show that both CELF and MBNL factors are required for regeneration in planarians *in vivo*, and that they act by antagonizing each other in the control of neoblast self-renewal and differentiation.

## Discussion

In this study we systematically characterized AS in planarian stem cells. We find widespread and strong differences in the pre-mRNA processing of genes expressed in both stem and differentiated cells. Most of these AS differences were evolutionarily conserved between two planarian species, strongly arguing for functional importance (*Irimia et al., 2009*). Furthermore, even several human genes undergo stem cell-specific AS in similar regions to those of their planarian orthologs. Moreover, we also found a large number of introns that are specifically retained in neoblast transcripts and that reduce the production of functional proteins in this cell type. Our results thus provide evidence that IR may play an important and distinct role in the regulation of stem cell gene expression.

Our results further indicate that BRULI and MBNL proteins functionally interplay to regulate planarian regeneration. *bruli* is highly expressed in stem cells and contributes to shape neoblast-specific transcriptomes at least in part through regulation of exon skipping and intron retention. Depletion of *bruli* results in neoblast loss and thus in a complete lack of regeneration. On the other hand, *mbnl* factors are more expressed in differentiated cells, where they contribute to establish differentiated gene expression. Therefore, loss of *mbnl* function is likely to affect neoblast differentiation, thereby reducing and/or slowing down regeneration. Our computational, biochemical and functional experiments indicate that CELF and MBNL proteins functionally antagonize by directly binding to their targets to control stem cell biology and regeneration in planarians (*Figure 8*). Thus, these two families of RBPs not only play antagonistic roles in the regulation of multiple mammalian differentiation systems (*Kalsotra et al., 2008*; *Lin et al., 2006*; *Ward et al., 2010*), but directly control stem cell-specific AS in an evolutionarily distant organism. Moreover, MBNL proteins have been shown to be major direct negative regulators of mammalian ESC-differential AS (*Han et al., 2013*; *Venables et al., 2013*), similarly to what we describe here for planarian neoblast AS. While MBNL targets are largely not shared between planarians and humans (only seven orthologous gene groups have MBNL-regulated, non-homologous, exons in both species), the upstream role of MBNL proteins as repressors of stem cell-differential AS appears equivalent in both species. This suggests that direct negative regulation of stem cell transcriptomes by MBNL proteins, and perhaps positively by

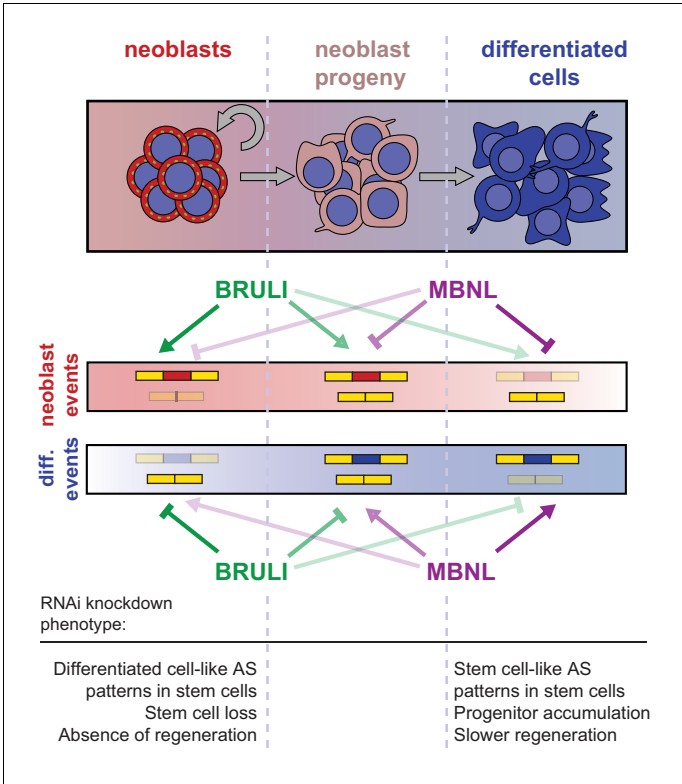

**Figure 8.** Model for BRULI and MBNL regulation of neoblast-specific AS. Schematic representation of BRULI and MBNL regulation of AS in different planarian cell fractions and respective RNAi mediated knockdown phenotypes.

CELF factors, is a deeply conserved ancestral feature of animal stem cells. Indeed, a recent study has shown that sponges – the most basal metazoan lineage –also differentially express a CELF homolog in their stem cells and an MBNL homolog in differentiated cells (*Alié et al., 2015*). However, since the true homology status of planarian, poriferan and human stem cells is unclear, it also possible that CELF and MBNL proteins form an ancestral developmental regulatory module that has been deployed to regulate pluripotent vs. differentiated transcriptomes multiple times across evolution. For instance, in *Drosophila melanogaster* a Bruno homolog (*aret*) is only highly expressed in the early pluripotent embryo stages (from 0 to 4 hr post fertilization) during development, while the embryonic expression of the single Mbnl ortholog (*mbl*) gradually increases from 6 hr post fertilization (*Graveley et al., 2011*). Furthermore, we have shown that not only the antagonistic expression but also the binding specificities of BRULI and MBNL proteins are very similar to those of their human counterparts, an observation that is in line with the high level of conservation of binding specificities and positional regulatory codes described for several other RBPs (*Brooks et al., 2011*; *Irimia et al., 2011*; *Lareau and Brenner, 2015*; *Ray et al., 2013*; *Wang et al., 2012*; *Loria et al., 2003*). The dual regulation by CELF and MBNL suggests that other interactions between factors in the control of AS might exist.

In summary, we have unveiled an ancient post-transcriptional antagonistic regulatory switch formed by CELF and MBNL factors that is associated with pluripotency in multiple animal systems. This evolutionary conservation contrasts with the apparent high turnaround of stem cell regulation at the transcription factor level, in which the core set of TFs that regulate mammalian ESCs is likely a lineage-specific innovation (*Fernandez-Tresguerres et al., 2010*). Future investigation of post-transcriptional regulation in other *in vivo* systems of stem biology and regeneration should further contribute to establishing ancestral mechanisms for the control of pluripotency in metazoans.

## Materials and methods

### Planarian genome annotation

We used the publically available assembly version 3.1 of the genome of the freshwater planarian *Schmidtea mediterranea*. Given the relatively fragmentary nature of this genome assembly, we used a recent *de novo* genome-independent transcriptome assembly based on multiple RNA-Seq data sources (Dresden transcriptome - http://planmine.mpi-cbg.de/) as transcript reference for the annotation. The 26,562 transcripts from this transcriptome were mapped to the genome using *blat* with the following parameters:–q = rna –fine (to search for initial and terminal exons in high quality transcripts). The output was then sorted using pslSort, and then filtered using pslCDnaFilter with the following parameters: -minCover = 0.75 -minId = 0.96 -globalNearBest = 0.005 –filterWeirdOverlapped. If multiple redundant mappings (i.e. of overlapping sequences in the query) of a single transcript fulfilled these conditions, only the hit with the best *blat* score was kept. If different regions of a transcript mapped to different scaffolds (936 transcripts), the transcript was split into two or more gene models, one in each scaffold (gene sub-models were designated with suffixes following '_2', '_3', etc.).

Next, a series of filters were applied to remove usual annotation errors. First, the strand of a gene model was flipped if there were more CT-AC than GT-AG introns (often occurring in gene chimeras in opposite orientations; 301 gene models). Second, extra multiple-mapping transcript regions within the same scaffold were filtered out (in 281 gene models). Third, exons separated by annotated introns shorter than 30 nucleotides (usually 1–5 nucleotides) were collapsed into single exons (these often reflected UTR polymorphisms or discrepancies between the transcriptome and genome assemblies; 7665 merged exons). This resulted in a GTF containing 23,161 single-transcript gene models. We next used TransDecoder (version 1) (*Haas et al., 2013*) with default parameters and compared against the Pfam A database to identify coding sequences (CDS). A total of 15,008 gene models had detected open reading frames (ORFs). For those cases in which TransDecoder detected more than one ORF in a given transcript (usually due to assembly errors that introduce PTCs), we generated multiple transcript identifiers (designated with suffixes following '_b', '_c', etc.). Custom perl scripts were then used to add start_codon and stop_codon lines to the GTF. Finally, to provide meaningful gene names, we performed tblastx (e-value 0.001, without low complexity filter) of the original planarian transcriptome against human mRNAs from Ensembl v71 and assign the gene symbols of best human hits (for 12,171 cases). This genome annotation was used as reference for our pipeline for alternative splicing (AS) identification, which includes multiple steps to identify non-annotated exons and splice sites (see below).

### Library preparation and RNA sequencing

Libraries for RNA sequencing generated in the present study were produced with poly-A selected RNA and according to the manufacturer's directions. All sequences were generated using Illumina HiSeq2000 machines in high yield mode. Read lengths, number of reads and percentage of reads mapped to the genome, as well accession number and sources for public RNA-Seq data are provided in *Figure 1—source data 1*.

### Genome-wide identification and quantification of alternative splicing

Identification and quantification of major types of AS events, including exon skipping (comprising events with single or multiple cassette exons and microexons between 3–27 nucleotides), intron retention (IR), and alternative 3' and 5' splice sites (Alt3 and Alt5) were performed using *vast-tools*, a recently described multi-module pipeline that has been applied to human, mouse and chicken (*Gueroussov et al., 2015*; *Irimia et al., 2014*). Here, we provide a summary of its main features and the specific modifications related to the planarian annotation.

For exon skipping (AltEx), we used three complementary approaches, as described in (*Irimia et al., 2014*). First, a 'transcript-based' approach uses full transcript information from multiple sources to identify single or groups of multiple cassette exons (consecutive alternative internal exons flanked by constitutive exons). Given the limited availability of Expressed Sequence Tags (ESTs) and cDNAs for planarians, we used other published *de novo* transcriptome assemblies as ESTs (*Adamidi et al., 2011*; *Blythe et al., 2010*; *Labbé et al., 2012*; *Onal et al., 2012*; *Resch et al.,*

2012; *Rouhana et al., 2012*). In addition, we also used RNA-Seq-based transcript annotations, derived from our multiple samples (*Figure 1—source data 1*) individually using STAR (*Dobin et al., 2013*) and cufflinks (*Trapnell et al., 2010*). Novel junctions obtained in the 'splice site-based' module (see below) were also incorporated. Finally, 215 additional cassette exons identified from previous transcriptomic assemblies (*Labbé et al., 2012*; *Onal et al., 2012*) were incorporated manually (indicated as 'CASSETTEd' in the Sme.EXSK.Template.*.txt files from *vast-tools*, see below). With this information, we identified individual or groups of neighbouring internal exons that are skipped in certain transcripts. Next, to quantify their inclusion levels, we mapped RNA-Seq reads to combinations of exon-exon junctions (EEJs) that define each splicing event. For the case of single exon skipping events, we generated EEJs for C1A, AC2 and C1C2 (A represents the alternative exon and C1 and C2 represent the neighboring constitutive exons), requiring a minimum of eight positions from each exon. For multi-exon events we generated all possible forward combinations from C1 to C2 exons. If multiple alternative 5′ and/or 3′ splice sites were associated with any alternative, C1 or C2 exons, they were also included in the combinations.

Second, a 'splice site-based' module utilizes the joining of all hypothetically-possible EEJ forward combinations from annotated and *de novo* splice sites (as described in (*Han et al., 2013*), where it was used to identify Embryonic Stem Cell [ESC]-differential exons). To identify splice sites *de novo*, for each annotated splice site donor/acceptor, we scanned two downstream/upstream introns for potential splice site acceptors/donors that would constitute a novel EEJ. Next, after subtracting the reads that map to the genome, we mapped our RNA-Seq data (*Figure 1—source data 1*) to this library of all potential EEJs, and considered 'novel splice sites' those supported by at least five reads mapped to multiple positions of the EEJ. Identification and quantification of cassette exons was done as described in (*Han et al., 2013*). Third, a 'microexon module' includes *de novo* searching of pairs of donor and acceptor splice sites in intronic sequence to detect novel, very short (i.e. 3–15 nt) microexons and subsequent quantification of inclusion levels using exon-microexon-exon junctions (EEEJ) (*Irimia et al., 2014*). The outputs from the three AltEx modules were combined to produce a non-redundant list of cassette exons and associated quantifications. For exons that are identified by more than one module, the representative with the highest overall read coverage is kept. In case of equal coverage, priority is given to events derived from the 'transcript-based' module, followed by those from the 'microexon' module. For intron retention (IR), we used the approach described in (*Braunschweig et al., 2014*). Finally, to detect and quantify Alt3 and Alt5 events, we used the output from mapping RNA-Seq reads to the EEJ library generated by the 'splice site-based module', which provides information on the usage of every hypothetical splice site donor and acceptor, as described in (*Irimia et al., 2014*). In all modules, quantification of alternative sequence inclusion in the transcripts is derived only from junction reads (either EEJs or EIJs). Raw RNA-seq reads were processed and EEJ/EIJ read counts corrected for mappability obtained as previously described (*Irimia et al., 2014*). These counts were used to derive alternative sequence inclusion levels using the 'Percent Splice In' metric (PSI; percent of transcripts from a given gene that include the alternative sequence).

The different modules to detect and quantify AS have been integrated into *vast-tools* (https://github.com/vastgroup/vast-tools; species key 'Sme'). Associated VASTDB files can be downloaded at http://vastdb.crg.eu/libs/vastdb.sme.31.1.15.tar.gz.

Finally, steady state mRNA levels for each gene were quantified using the metric 'corrected (for mappability) Reads Per Kilobase pair per Million mapped reads' (cRPKMs, [*Labbé et al., 2012*]) using the original *de novo* transcriptome employed for genome annotation (Dresden transcriptome - http://planmine.mpi-cbg.de/), and it is also implemented in *vast-tools* (option –expr in *vast-tools* align).

## Alternative splicing definition and minimum read coverage

For all types of events, we used the same definition for a given sequence to be considered alternatively spliced: $10 \leq PSI/PSU/PIR \leq 90$ in at least 10% of the samples with sufficient read coverage and/or a range of PSI/PSU/PIRs $\geq 25$ across all samples with sufficient read coverage. A given event was considered to have sufficient read coverage in a particular RNA-Seq sample according to the following criteria:

- For AltEx (except for those quantified using the microexon pipeline): (i) $\geq 10$ actual reads (i.e. before mappability correction) mapping to the sum of exclusion EEJs, **OR** (ii) $\geq 10$ actual reads mapping to one of the two inclusion EEJs, and $\geq 5$ to the other inclusion EEJ.

- For microexons: (i) $\geq 10$ actual reads mapping to the sum of exclusion EEJs, **OR** (ii) $\geq 10$ actual reads mapping to the sum of inclusion EEEJs.

- For IR: (i) $\geq 10$ actual reads mapping to the sum of skipping EEJs, **OR** (ii) $\geq 10$ actual reads mapping to one of the two inclusion EIJs, and $\geq 5$ to the other inclusion EIJ.

- For Alt3 and Alt5: $\geq 10$ actual reads mapping to the sum of all EEJs involved in the specific event.

The total number of AS events by type identified using all the RNA-Seq data described above are depicted in *Figure 1—figure supplement 2*.

## Definition of neoblast-differential alternative splicing

To identify AS events that were differentially regulated between X1 and Xins cell fractions, we first required that any given AS event has sufficient read coverage (see above) in a minimum of 3 out of 6 X1 and 3 out of 8 differentiated samples (either Xins or neoblast-depleted whole worms; referred below as 'diff'). We implemented a set of non-mutually exclusive definitions to maximize the detection of different patterns of neoblast AS regulation, including both quantitative (i.e. with large PSI differences between the samples from the X1 and diff groups) and qualitative (i.e. in which one of the isoforms is present only in the set of X1 or differentiated cell samples, even if with relatively small PSI differences between both groups). The following definitions were employed:

1. absolute $(\text{Mean}_{X1} - \text{Mean}_{diff}) \leq 25$ **AND**
   [$\text{Range}_{X1} <$ absolute $(\text{Mean}_{X1} - \text{Mean}_{diff})/2$ **OR**
   $\text{Range}_{diff} <$ absolute $(\text{Mean}_{X1} - \text{Mean}_{diff})/2$ **OR**
   $\text{SD}_{X1} <$ absolute $(\text{Mean}_{X1} - \text{Mean}_{diff})/2$ **OR**
   $\text{SD}_{diff} <$ absolute $(\text{Mean}_{X1} - \text{Mean}_{diff})/$**2**]
2. absolute $(\text{Mean}_{X1} - \text{Mean80}_{diff}) \leq 20$ **AND**
   [$\text{SD}_{X1} <$ absolute $(\text{Mean}_{X1} - \text{Mean}_{diff})/4$ **OR**
   $\text{SD}_{diff} <$ absolute $(\text{Mean}_{X1} - \text{Mean}_{diff})/$**4**]
3. $\text{Total\_samples}_{diff} \geq 4$ **AND**
   $\text{Total\_samples}_{X1} \geq 4$ **AND**
   [($\text{Min}_{X1} \geq 98$ **AND** $\text{Max}_{diff} < 90$) **OR**
   ($\text{Max}_{X1} \leq 2$ **AND** $\text{Min}_{diff} > 10$) **OR**
   ($\text{Min}_{diff} \geq 98$ **AND** $\text{Max}_{X1} < 90$) **OR**
   ($\text{Max}_{diff} \leq 2$ **AND** $\text{Min}_{X1} > 10$)]

where $\text{Total\_samples}_{X1/diff}$ is the number of X1/differentiated samples with enough read coverage; $\text{Mean}_{X1/diff}$ is the mean PSI/PSU/PIR for all X1/differentiated samples; $\text{Mean80}_{diff}$ are the mean PSI/PSU/PIR for the differentiated samples excluding the 20% of samples with the most distant PSI/PSU/PIRs from the X1 mean value (i.e. ninth to 10th deciles); $\text{Min}_{X1/diff}$ is the minimum PSI/PSU/PIR value for X1/differentiated samples; $\text{Max}_{X1/diff}$ is the maximum PSI/PSU/PIR value for X1/differentiated samples; $\text{Range}_{X1/diff}$ is the difference between $\text{Max}_{X1/diff}$ and $\text{Min}_{X1/diff}$; and $\text{SD}_{X1/diff}$ is the standard deviation of PSI/PSU/PIR values for X1/differentiated samples. In addition, we required bona fide neoblast-differential AS events to have a $p<0.05$ when comparing X1 and differentiated samples using the B-statistic, i.e. the empirical Bayes log-odds of differential PSI/PSU/PIRs (*Smyth, 2004*) (as implemented in 'ebayes', from the limma package in R). AS events not differentially regulated between X1 and differentiated cells ('AS_nonX1' events) were defined as those AS events (as defined in the section 'Alternative splicing definition and minimum read coverage') that had (i) enough read coverage in at least 3 X1 samples and 3 differentiated samples, (ii) $|(\text{Mean}_{X1} - \text{Mean}_{diff})| < 10$, and (iii) $p \geq 0.05$ on the Bayesian test.

## Gene Ontology and KEGG enrichment analysis

GO terms were mapped and extracted from an InterProScan (*Zdobnov and Apweiler, 2001*) annotation. Enrichment was performed using the library GOstats (*Falcon and Gentleman, 2007*) (testDirection = 'over', conditional = TRUE). The background was defined as all expressed transcripts (log2 (TPM+1)>1.5) in all samples any given experiment. For time course experiments the test set was

defined as the union of downregulated transcripts in days 20, 25 and 30 (threshold log2 FC <−0.7). KEGG annotation was performed using the KEGG Automatic Annotation Server (*Moriya et al., 2007*) (KAAS) with single best hit and a custom set of species (*Homo sapiens* (hsa), *Mus musculus* (mmu), *Gallus gallus* (gga), *Xenopus laevis* (xla), *Danio rerio* (dre), *Strongylocentrotus purpuratus* (spu), *Drosophila melanogaster* (dme), *Caenorhabditis elegans* (cel), *Helobdella robusta* (hro), *Lottia gigantea* (lgi), *Schistosoma mansoni* (smm), *Nematostella vectensis* (nve), *Amphimedon queenslandica* (aqu), *Saccharomyces cerevisiae* (sce)). KEGG enrichment was assessed using the same subsets as in the cluster enrichment analyses and using the GOstats library (testDirection = 'over', conditional = TRUE).

## Comprehensive analysis of RNA-binding protein effects on AS

We annotated RNA binding proteins (RBPs) in our reference transcriptome. RBPs were identified using the InterProScan annotation by filtering PFAM (*Finn et al., 2014*) ids associated with known RNA binding domains [RRM (PF05172, PF08675, PF00076, PF04059, PF08777, PF13893, PF14259, PF03467, PF03468, PF10598, SM00360, PS50102); KH (PF00013, PF07650, SM00322); helicases (PF00270) and other domains (PF12171, PF05741, PF12251, PF14709, PF00035, PF02037, PF00806, PF01423, PF02171, PF12701, PF14438)] and top BLAST (*Camacho et al., 2009*) hits against complete proteomes (Uniprot [*UniProt 2015*]) of several species (Human, mouse, *C. elegans*, fruitfly, *S. mansoni*, *Dugesia japonica*, zebrafish, chicken and xenopus) filtered for relevant key terms (RNA-binding, RNA, binding). From this list, a shortlist of candidate AS factors known to regulate AS in other organisms were selected (*Figure 4—source data 1*) and those with the highest evidence were selected for RNAi studies (*Figure 4—source data 1*, colored). We grouped those in clusters of orthology using KEGG annotations. When the X1/Xins ratio of different orthologs of the same group differed greatly (indicating differential expression of the orthologs), the group was split into two groups to separate the orthologues highly specifically expressed in X1 from the highly specifically expressed in Xins (*Figure 4—source data 3*). Since in a previous version of the transcriptome (BIMSB [*Adamidi et al., 2011*]) the mbnl-1 locus had two different transcripts IDs (isotig19687 and isotig20952), one dsRNA was designed against both of them (noted as *mbnl-1* and *mbnl-1\**, *Supplementary file 1*).

Comparison to spliceosomal components was done to gain insight of housekeeping gene expression in stem cells vs. differentiated cells. We selected core spliceosomal components from Complexes A, B and C by searching the relevant KEGG terms in our KEGG annotation. When two or more paralogs with the same KEGG term were found those were removed since their expression patterns were frequently more divergent, indicating possible subfunctionalization of paralog genes. The final list of transcripts used and their expression values is in *Figure 4—source data 3*.

RNAi was done as described below. Regeneration tests were performed 22 days after RNAi treatment. RNAs from FACS samples and whole worm samples for sequencing were extracted 10 days after RNAi treatment as described below. Since only one replicate was performed or this experiment, we used a more stringent minimum read coverage:

- For AltEx (except for those quantified using the microexon pipeline): (i) ≥15 actual reads (i.e. before mappability correction) mapping to the sum of exclusion EEJs, OR (ii) ≥15 actual reads mapping to one of the two inclusion EEJs, and ≥10 to the other inclusion EEJ.

- For microexons: (i) ≥15 actual reads mapping to the sum of exclusion EEJs, OR (ii) ≥15 actual reads mapping to the sum of inclusion EEEJs.

For each AS factor group, we quantified the percent of X1-differential alternative exons with sufficient read coverage across the 8 RNAi treated samples (n = 173) that showed a ΔPSI ≥15 between the treated and the control samples towards the PSI in X1.

## Prediction of the impact of AS events on proteomes

Alternative sequences were first mapped to coding (CDS) or non-coding (UTRs) sequences based on our TransDecoder-based genome annotations. For those alternative sequences that are not present in our reference annotation, mapping was projected based on the information of the upstream exon. Then, sequences that contained in-frame stop codons (based on annotations and in-frame sequence translation from the upstream exon) or start codons (based on annotations) were flagged. Following a recent study (*Irimia et al., 2014*), AS events were predicted to generate alternative

protein (ORF-preserving) isoforms upon both inclusion and exclusion when they fall in the CDS, do not disrupt the reading frame (i.e. they have lengths multiple of three nucleotides), and do not contain in-frame stop codons predicted to trigger non-sense mediated decay (NMD; those stop codons that fall further than 50 nucleotides away from a downstream EEJ). Furthermore, if an alternative sequence is predicted to introduce an in-frame stop codon that would not trigger NMD but would generate a protein >100 aminoacids shorter than the annotated isoform, it was also considered as an ORF-disrupting event. For multiexonic cassette events (arrays of more than one alternative exon), NMD/ORF-disruption was assessed for all exons as a group based on their inclusion patterns in neoblast and differentiated samples (e.g. if one frame-shifting exon is downregulated in neoblast samples and another upregulated, both the neoblast and differentiated isoforms can have intact reading frames). For ORF-disrupting AS events, two main categories were defined: ORF-disruption upon sequence inclusion or upon sequence exclusion. Therefore, depending on the neoblast inclusion pattern, we defined events that cause ORF disruption in neoblast samples, but not in other tissues ('ORF disruption in X1'), or the opposite, AS events that disrupt ORFs only in differentiated samples ('ORF disruption in Xins').

To investigate the overlap of alternative exons with disordered regions and protein domains we first mapped the alternative exons as well as the upstream (C1) and downstream (C2) exons to the proteome as annotated in our genome annotation (see above). Next, to include unannotated exons, we recreated novel protein isoforms by introducing the exonic sequence downstream of the upstream C1 exon. The final set of proteins was used to run *de novo* predictions using Disopred2 (for structural disorder, [*Ward et al., 2004*]), and Pfam (for protein domains, (*Finn et al., 2014*); only domains from the A module were used), both with default parameters. For both disorder and Pfam domains, the fraction of residues from alternative exons overlapping those features are reported. For consistency, for all exon classes, only exons that would generate protein isoforms both when included and skipped (i.e. internal exons with lengths multiple of three nucleotides without in-frame stop codons) were analyzed.

## Study of evolutionary conservation in *D. japonica*

With the aim of designing *D. japonica*-specific primers for all validated neoblast-differential AS events (*Figure 1—figure supplement 4*), we first blasted the *S. mediterranea* alternative sequences, as well as the neighbouring upstream and downstream exons against a recently published transcriptome assembly of *D. japonica*. If at least two of these sequences mapped against the same contig as the best hit, this was designated as the best ortholog and primers were designed in equivalent *D. japonica* sequences. Next, for those cases that we could not match using this strategy, we blasted the translated *S. mediterranea* full protein sequence again the *D. japonica* transcriptome using tblastn. The best *D. japonica* hit was considered as the potential ortholog, and further manual annotation through exon-specific alignments was performed to identify the region homologous to the *S. mediterranea* AS event. All primer *D. japonica* sequences are provided in *Supplementary file 1*.

## Conservation of neoblast-differential and human ESC-differential AS events

We used two approaches to assign orthology between planarian and human genes with neoblast-differential and ESC-differential, respectively. First, Inparanoid v8.0 (*Ostlund et al., 2010*) was run for one reference protein per gene in planarian and human using default parameters. If the planarian and human proteins were grouped in the same cluster, they were considered putative orthologs. Second, we blasted the planarian proteins against the human reference proteome and the first three hits were matched against the subset of genes with ESC-differential exons. From these two complementary approaches, orthology calls were manually verified and dubious mis-assignments were discarded. In total 15 orthologous groups comprising 17 planarian genes (with neoblast-differential 30 exons) and 20 human genes (harbouring 22 ESC-differential exons). Next, to assess orthology at the exonic level, we mapped intron positions into protein sequence alignments, as previously described (*D'Aniello et al., 2008*); in addition, conserved protein domains were used as milestones to define homologous protein regions (*Figure 3—source data 2*). With this information, we evaluated whether (i) the exons fall in equivalent protein regions (e.g. within two protein domains), (ii) the sequence was orthologous, and (iii) the exons were orthologous (based on the intron position alignments). The

exons were defined to be in 'similar protein regions' (*Figure 3—source data 1*) if they fall in the same relative region of the protein (as per (i)) and the distance in the alignment is lower than 100 residues.

## RNAcompete and motif enrichment analyses

The Full-length version of *bruli*, *mbnl-1* (both the X1- and Xins-enriched isoforms), *mbnl-like-1* and *mbnl-like-2* were cloned into GST-tagged vector (pTH6838) using primers listed in *Supplementary file 1*. GST-tagged proteins were expressed and purified from *Escherichia coli* as previously described (*Ray et al., 2009*, *2013*).The RNA pool generation, RNAcompete pull-down assays, and microarray hybridizations were performed as previously described (*Ray et al., 2009*, *2013*). Briefly, GST-tagged RBPs (20 pmoles) and RNA pool (1.5 nmoles) were incubated in 1 mL of Binding Buffer (20 mM HEPES pH 7.8, 80 mM KCl, 20 mM NaCl, 10% glycerol, 2 mM DTT, 0.1 mg/mL BSA) containing 20 mL glutathione sepharose 4B (GE Healthcare) beads (pre-washed 3 times in Binding Buffer) for 30 min at 4°C, and subsequently washed four times for two minutes with Binding Buffer at 4°C. One-sided Z-scores were calculated for the motifs as described previously (*Ray et al., 2013*).

## Motif analysis

For the alternatively spliced exons from each of the test sets (I-IV in *Figure 5*) and corresponding background sets ('AS_nonX1' events [see above] that showed a ΔPSI < 5 between *bruli* or *mbnl* KD and control), the sequences of the flanking introns, as well as the alternative exons themselves, were fetched from the planarian genome. The sub-sequences corresponding to the first and last 80nt of each intron were further subdivided into 20-nt bins. Introns shorter than 160-nt were excluded from the analysis (see below). Next, all 7-mer occurrences within each bin were counted and summed for corresponding bins over all events in the signal and background sets. We then selected the top 10% 7-mers with the highest affinity for an RNA-binding protein, as measured by RNAcompete (z-score). For each bin, we computed the sum-total occurrences of the high affinity 7-mers in the signal and control events. If enrichment was observed, a P-value was calculated using Fisher's exact test. If significance was reached (p<0.01) the bin is represented as a filled rectangle, with the saturation of the color proportional to $-\log_{10}$(P-value). Non-significant bins are represented as empty rectangles. Similar results were obtained when 80 nucleotides upstream and downstream the exon were assayed, without discarding shorter introns.

To calculate enrichment of general RBPs in sequences associated with X1-included and X1-excluded exons (*Figure 4—figure supplement 1*), we used the entire CISBP-RNA database (*Ray et al., 2013*) and performed a similar analysis for each of the available RBP records, using all 'AS_nonX1' events as background. We sorted the RBPs by the strongest significance in any of the bins and employed a cutoff of $p<10^{-4}$ to restrict the output to the most significant hits.

## Animals and RNAi treatments

All animals belonged to the Berlin-1 strain of asexual type *Schmidtea mediterranea*, recently generated from one single individual. For RNAi experiments, dsRNAs were synthesized as previously described (*Solana, 2013*). Animals were injected with dsRNA against the coding region of the gene of interest (*control*(*RNAi*) planarians were injected with dsRNA coding for GFP) for three consecutive days (days one, two and three after RNAi) and kept at 20°C, as previously described. dsRNAs were delivered at a concentration of 1 μg/μl. When multiple dsRNAs were used simultaneously, each dsRNA was injected at a concentration of 1 μg/μl in the same solution. Since in a previous version of the transcriptome (BIMSB) the *mbnl-1* locus had two different transcripts IDs (isotig19687 and isotig20952), one dsRNA was designed against both of them (*Supplementary file 1*). When appropriate, the concentration of negative control dsRNA coding for *gfp* was adjusted in the control samples to the maximum concentration of dsRNA injected in the experimental groups.

## FACS experiments

FACS experiments were performed as previously described (*Hayashi et al., 2006*; *Onal et al., 2012*). Essentially, planarians were cut into little pieces on ice and in the presence of trypsin to help cell dissociation. Cells were then sequentially filtered through 40 μm and 20 μm filters and stained

with the cytoplasmic dye Calcein-AM (BD Biosciences, at a final concentration of 0.5 µg/ml) and the nuclear dye Hoechst 33,342 (Fluka Biochemika, at a final concentration of 20 µg/ml). Propidium Iodide was used to discard dead cells. Cells were then sorted with a BD FACSAria III directly into Trizol LS containing tubes. X1 gating is achieved by sorting the population of cells with double content of DNA. From the single content, those cells with low calcein staining were gated as X2 and those with high calcein staining were gated as Xins. Control extractions with irradiated planarians were used to assist correct gating. RNA was extracted using Trizol LS (Ambion) following manufacturer instructions. For *Dugesia japonica* experiments, minor adjustments in the nuclear staining and the FACS gating were applied since this species has a larger DNA content (*Nishimura et al., 2015*).

## RT-PCR, qPCR and *in situ* hybridization

For RT-PCR and qPCR analyses, RNA was extracted with self-made Trizol reagent (modified from [*Chomczynski and Sacchi, 1987*]) reverse transcribed with an oligodT primer using Maxima H Minus Reverse Transcriptase (Thermo Scientific, Waltham, MA). RT-PCRs were visualized in 2.5–3.3% agarose gels. Relative ratios between the two isoforms were then calculated based on the relative intensity of the PCR bands, measured using Image J. qPCR experiments were technically replicated twice, and performed with 2 biological replicates of each condition. Each sample was always loaded in triplicates. *In situ* hybridization was performed as previously described (*King and Newmark, 2013*) using an Intavis Vsi Pro robot. Probes were synthesized from PCR amplicons as previously described (*Solana, 2013*). All primers used in each experiment and provided in *Supplementary file 1*.

## Regeneration speed tests

For regeneration tests, 10 planarians per time point and condition were selected, cut at various time points, and observed and scored daily under the scope. At each scoring time point, identification of normally looking eyespots was used as proxy for complete regeneration. Trunk (amputated head and tail) and trunk/tail (only head was amputated) pieces were observed for 10 days, tail pieces were observed for 12 days as their regeneration time is slower. All control trunk/tail, trunk and tail pieces used in this study had visible eyespots by day 6, 7 and 10 respectively.

## Transcriptomic analysis of knockdown experiments

RNA-Seq reads were processed and filtered for low quality and 3′ and 5′ adapter removal using Flexbar v2.5 (*Dodt et al., 2012*). in-silico ribosomal depletion was performed with bowtie2 (*Langmead and Salzberg, 2012*) in local mode (bowtie2 –local –very-sensitive-local -x rRNA_INDEX -p 8 -U FASTQ –un FILTERED.fastq) against a pool of platyhelminthes rRNA index. Filtered reads were then mapped to the reference planarian transcriptome using bowtie2 default parameters. Transcript quantification was performed using htseq-count (*Anders et al., 2015*).

To assess differential expression we resorted to previously described clustering of genes by gene expression profiles in FACS sorted populations (*Onal et al., 2012*).Transcripts from the original source were first mapped to our reference transcriptome using BLAT (*Kent, 2002*) and pslReps with the options -minCover = 0.5 and -minAli = 0.1 -nearTop = 0.005 filtering parameters. The clusters were assigned by matching 1-to-many. In case of multiplicity, the following rules were applied: if a transcript was matched to cluster 1 and 2 or 5 and 6, cluster 2 and cluster 6 were assigned, respectively. Other combinations were considered ambiguous and discarded and the cluster left blank. Next, the control samples were normalized using quantile normalization with the function normalize. quantiles (R-base library - http://www.R-project.org) to reduce biases by rRNA contamination. The dataset was then filtered for reliably detected transcripts using a threshold of $log2(TPM+1)>1.5$ laying a total of 12,599 transcripts. Two conditional sets (*bruli* KD downregulated, *mbnl* KD downregulated) were defined as transcripts below a threshold of $-0.7$ log2 FC over the corresponding control sample. Differential enrichment of downregulated genes from specific clusters compared to the global transcriptome was assessed using hypergeometric test, using all expressed transcripts (12,599) as background and performing multiple testing Bonferroni correction on the p-value.

## Acknowledgements

We thank all members of the Rajewsky lab for helpful discussion and Johanna Schmid for technical assistance. JS and NR acknowledge funding from the Deutsche Forschungsgemeinschaft (SO 1308/2-1). We acknowledge support of the Spanish Ministry of Economy and Competitiveness, 'Centro de Excelencia Severo Ochoa 2013-2017', SEV-2012-0208 and BFU2014-55076-P to MI and a CIHR Operating Grant MOP-125894 to QDM and TRH.

## Additional information

### Competing interests

BJB: Reviewing editor, *eLife*. The other authors declare that no competing interests exist.

### Funding

| Funder | Grant reference number | Author |
|---|---|---|
| Deutsche Forschungsge-meinschaft | SO 1308/2-1 | Jordi Solana |
| Ministerio de Economía y Competitividad | BFU2014-55076-P | Manuel Irimia |
| Canadian Institutes of Health Research | MOP-125894 | Quaid Morris Timothy R Hughes |
| Ministerio de Economía y Competitividad | SEV-2012-0208 | Manuel Irimia |

The funders had no role in study design, data collection and interpretation, or the decision to submit the work for publication.

### Author contributions

JS, MI, Conception and design, Acquisition of data, Analysis and interpretation of data, Drafting or revising the article; SA, Acquisition of data; MRO, BJB, Analysis and interpretation of data, Drafting or revising the article; VZ, Acquisition of data, Drafting or revising the article; MJ, JT, Analysis and interpretation of data; DR, QM, TRH, Acquisition of data, Analysis and interpretation of data, Drafting or revising the article; NR, Conception and design, Analysis and interpretation of data, Drafting or revising the article

### Author ORCIDs

Jordi Solana, http://orcid.org/0000-0002-6770-3929
Manuel Irimia, http://orcid.org/0000-0002-2179-2567
Nikolaus Rajewsky, http://orcid.org/0000-0002-4785-4332

## Additional files

### Supplementary files

• Supplementary file 1. List of primers for RT-PCR AS events, RNA compete, dsRNA and probes and qPCRs.

### Major datasets

The following datasets were generated:

| Author(s) | Year | Dataset title | Dataset URL | Database, license, and accessibility information |
|---|---|---|---|---|
| Solana J, Irimia M, Orejuela MR, Rajewsky N | 2016 | X1, X2 and Xins fractions and whole woms in wild type, control(RNAi), bruli(RNAi), mbnl(RNAi), and other splicing factors | http://www.ncbi.nlm.nih.gov/geo/query/acc.cgi?acc=GSE75594 | Publicly available at NCBI Gene Expression Omnibus (accession no: GSE75594) |

| | | | | |
|---|---|---|---|---|
| Solana J, Irimia M, Orejuela MR, Rajewsky N | 2016 | RNAcompete data | http://www.ncbi.nlm.nih.gov/geo/query/acc.cgi?acc=GSE75554 | Publicly available at NCBI Gene Expression Omnibus (accession no: GSE75554) |
| Solana J, Irimia M, Ayoub S, Orejuela MR, Zywitza V, Jens M, Ray D, Morris QD, Hughes TR, Blencowe BJ, Rajewsky N | 2016 | Time course of whole worms for phenotypic study of bruli(RNAi) and mbnl(RNAi) | http://www.ncbi.nlm.nih.gov/geo/query/acc.cgi?acc=GSE75296 | Publicly available at NCBI Gene Expression Omnibus (accession no: GSE75296) |

The following previously published datasets were used:

| Author(s) | Year | Dataset title | Dataset URL | Database, license, and accessibility information |
|---|---|---|---|---|
| Gene Expression Omnibus (GEO) | 2012 | Stem Cells; Schmidtea mediterranea; RNA-Seq | http://trace.ncbi.nlm.nih.gov/Traces/sra/?run=SRR496276 | Publicly available at NCBI Sequence Read Archive (accession no: SRR496276) |
| Whitehead Institute | 2015 | X1 transcriptome assembly | http://www.ncbi.nlm.nih.gov/sra/SRR1302103 | Publicly available at NCBI Sequence Read Archive (accession no: SRR1302103) |
| Gene Expression Omnibus (GEO) | 2012 | Stem Cell Progeny; Schmidtea mediterranea; RNA-Seq | http://trace.ncbi.nlm.nih.gov/Traces/sra/?run=SRR496278 | Publicly available at NCBI Sequence Read Archive (accession no: SRR496278) |
| HHMI | 2013 | RNA sequencing for transcriptome assembly - Smed Sexual(S2F2) biotype | http://www.ncbi.nlm.nih.gov/sra/SRR955511 | Publicly available at NCBI Sequence Read Archive (accession no: SRR955511) |
| HHMI | 2013 | RNA sequencing for transcriptome assembly - Smed Asexual(CIW4) biotype | http://www.ncbi.nlm.nih.gov/sra/SRR955099 | Publicly available at NCBI Sequence Read Archive (accession no: SRR955099) |
| Gene Expression Omnibus (GEO) | 2013 | control(RNAi) animals (unc22) 1; Schmidtea mediterranea; RNA-Seq | http://trace.ncbi.nlm.nih.gov/Traces/sra/?run=SRR867386 | Publicly available at NCBI Sequence Read Archive (accession no: SRR867386) |
| Gene Expression Omnibus (GEO) | 2012 | GSM929780: Differentiated tissues; Schmidtea mediterranea; RNA-Seq | http://trace.ncbi.nlm.nih.gov/Traces/sra/?run=SRR496280 | Publicly available at NCBI Sequence Read Archive (accession no: SRR496280) |
| UNIVERSITY OF ILLINOIS, URBANA-CHAMPAIGN | 2012 | Schmidtea mediterranea Control1 (for PRMT5) | http://trace.ncbi.nlm.nih.gov/Traces/sra/?run=SRR390350 | Publicly available at NCBI Sequence Read Archive (accession no: SRR390350) |
| UNIVERSITY OF ILLINOIS, URBANA-CHAMPAIGN | 2012 | Schmidtea mediterranea PRMT5 (RNAi)-1 | http://trace.ncbi.nlm.nih.gov/Traces/sra/?run=SRR390352 | Publicly available at NCBI Sequence Read Archive (accession no: SRR390352) |
| Gene Expression Omnibus (GEO) | 2012 | Stem Cells; Schmidtea mediterranea; RNA-Seq | http://trace.ncbi.nlm.nih.gov/Traces/sra/?run=SRR496277 | Publicly available at NCBI Sequence Read Archive (accession no: SRR496277) |
| Gene Expression Omnibus (GEO) | 2012 | Stem Cell Progeny; Schmidtea mediterranea; RNA-Seq | http://trace.ncbi.nlm.nih.gov/Traces/sra/?run=SRR496279 | Publicly available at NCBI Sequence Read Archive (accession no: SRR496279) |

| | | | | |
|---|---|---|---|---|
| Gene Expression Omnibus (GEO) | 2013 | control(RNAi) animals (unc22) 1; Schmidtea mediterranea; RNA-Seq | http://trace.ncbi.nlm.nih.gov/Traces/sra/?run=SRR867387 | Publicly available at NCBI Sequence Read Archive (accession no: SRR867387) |
| Gene Expression Omnibus (GEO) | 2013 | control(RNAi) animals (unc22) 1; Schmidtea mediterranea; RNA-Seq | http://trace.ncbi.nlm.nih.gov/Traces/sra/?run=SRR867388 | Publicly available at NCBI Sequence Read Archive (accession no: SRR867388) |
| Gene Expression Omnibus (GEO) | 2012 | GSM929780: Differentiated tissues; Schmidtea mediterranea; RNA-Seq | http://trace.ncbi.nlm.nih.gov/Traces/sra/?run=SRR496281 | Publicly available at NCBI Sequence Read Archive (accession no: SRR496281) |
| UNIVERSITY OF IL-LINOIS, URBANA-CHAMPAIGN | 2012 | Schmidtea mediterranea Control1 (for PRMT5) | http://trace.ncbi.nlm.nih.gov/Traces/sra/?run=SRR390351 | Publicly available at NCBI Sequence Read Archive (accession no: SRR390351) |
| UNIVERSITY OF IL-LINOIS, URBANA-CHAMPAIGN | 2012 | Schmidtea mediterranea PRMT5 (RNAi)-1 | http://trace.ncbi.nlm.nih.gov/Traces/sra/?run=SRR390353 | Publicly available at NCBI Sequence Read Archive (accession no: SRR390353) |

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
