## [Decision Letter]

Thank you for submitting your article "Conserved functional antagonism between CELF and MBNL proteins regulates stem cell-specific alternative splicing" for consideration by *eLife*. Your article has been favorably evaluated by Aviv Regev as the Senior editor and three reviewers, including Jernej Ule (Reviewer #2) and Alejandro Sánchez Alvarado (Reviewer #1), who is a member of our Board of Reviewing Editors. The following individual involved in the review of your submission has agreed to reveal their identity:

The reviewers have discussed the reviews with one another and the Reviewing Editor has drafted this decision to help you prepare a revised submission.

Summary:

In this comprehensive study, Dr. Solana et al. investigate the role that alternative splicing (AS) may or may not play in the regulation of stem cells *in vivo* using the planarian model system *S. mediterranea*. The authors report on the conservation of an AS program comprising hundreds of alternative exons, microexons and intron retention events that is differentially regulated. Additionally, the authors identify a cohort of RNA binding proteins associated with the regulation of the uncovered AS program. Chiefly, the authors focus on two known functional antagonists involved in AS regulation, i.e., CELF and MBNL. Knockdown of either CELF or MBNL factors lead to abnormal regeneration by affecting the regulation of self-renewal and differentiation sets of genes. Additionally, the authors find neoblast-unique patterns of splicing that is opposingly regulated by the *bruli* and *mbnl* genes in neoblasts versus differentiated cells. Overall, the experiments are carefully executed and the data produced opens a significant number of novel lines of future investigations.

Essential revisions:

The following issues need to be addressed fully in a revised submission.

1) The comparison of the planarian dataset to ES cells is somewhat superficial and could be improved if the authors were to delve in more detail on the role of MBNL in mammalian cells and its significance in regulating AS and relate it to the reported planarian data more specifically. For example, the fourth paragraph of the Introduction is confusing (*mbnl* promotes ES form of FoxP?). Also in the fourth paragraph of the Introduction: differentiated cell role is very relevant –therefore noting more detail would be helpful (which cell types were examined for instance).

2) Because a major emphasis of this paper lies on the evolutionary conservation of the CELF/MBNL AS regulation axis, the authors should attempt to determine whether there is or there is not conservation of the planarian regulated isoforms in other species, i.e., human and/or mouse stem cells. Is there any overlap between MBNL alternative spliced isoforms in human and the orthologous planarian genes in stem cells? For instance in the subsection “Unexpected abundance of neoblast-specific retained introns”, the authors put forward a very interesting mechanism that is, nonetheless, speculative. The authors could word this more conservatively by mentioning ES data, then state these genes are good targets to test in planarian neoblasts to test for mechanistic similarity. Overall, given the embryonic strategy used by mammalian embryos to derive ES cells from embryos undergoing implantation, and that such an attribute is unlikely associated with the common ancestor of planarians and mammals, the homology between ES cells and neoblasts is dubious. Nonetheless, opposing roles for CELF/MBNL factors in stem/progenitor/embryonic cells, or cells associated with multi/pluripotency in their last common ancestor is certainly possible and would be very interesting. Such a role could have been variously recruited to embryonic cells and different stem cell classes in the course of evolution. This could be assessed by studying many stem cell/embryonic cell classes in other organisms/animal tissues. It is also possible that opposing roles in AS programs for CELF/MBNL proteins is what is conserved and that this could be variously utilized in different types of biology. We recommend a more careful/nuanced wording throughout, but particularly in the discussion. This may help bring the evolutionary line of argumentation into sharper focus.

3) The model put forward in Figure 8 overinterprets the data. We find no clear evidence in the manuscript for "impaired differentiation". Additionally, the significance of the connecting lines and arrows is far from clear. Do these arrows represent phenotypes caused by RNAi loss of gene function or do they represent gene function? In other words, in Figure 8, bottom image, the arrow seems to imply that *bruli* functions to negatively regulate stem cell renewal, etc., but the proposed roles for *bruli* and *mbnl* are opposite to arrows. If the negative arrows are meant to reflect phenotype, that is an atypical and confusing way to summarize gene function. *mbnl* arrows reflect more interpretation than is possible from the data-the authors have not shown whether there is more progression from SC to prog cells or less progression from prog cells to differentiated cells.

4) The high conservation of tissue-specific splicing regulators and their regulatory codes has also been documented before (PMID: 21389270, 25576366, 20921232). The human CELF4 can rescue the function of *C. elegans* orthologue (PMID: 12906792) and the MBNL sequence specificity, its regulatory program and many of its binding sites are conserved between *Drosophila* and human (PMID: 22901804). The authors should describe this further in the manuscript.

5) Please confirm that all large scale transcription datasets are deposited in a publicly accessible repository.

---

## [Author Response]

Essential revisions:

*The following issues need to be addressed fully in a revised submission.*

1) The comparison of the planarian dataset to ES cells is somewhat superficial and could be improved if the authors were to delve in more detail on the role of MBNL in mammalian cells and its significance in regulating AS and relate it to the reported planarian data more specifically. For example, the fourth paragraph of the Introduction is confusing (mbnl promotes ES form of FoxP?). Also in the fourth paragraph of the Introduction: differentiated cell role is very relevant – therefore noting more detail would be helpful (which cell types were examined for instance).

The knockdown experiments of Han et al. were done in transformed cell cultures, but a large number of differentiated tissues (>15) and transformed and primary cell lines were included in their bioinformatic analyses. We have now made these details explicit in the text. In addition, we have edited parts of the Introduction to clarify the role of MBNL in repressing the ES isoform of FoxP1 and how this isoform regulates mammalian ESC biology. Finally, we have further expanded the discussion of the conservation of MBNL between planarian and mammalian stem cells (see below).

2) Because a major emphasis of this paper lies on the evolutionary conservation of the CELF/MBNL AS regulation axis, the authors should attempt to determine whether there is or there is not conservation of the planarian regulated isoforms in other species, i.e., human and/or mouse stem cells. Is there any overlap between MBNL alternative spliced isoforms in human and the orthologous planarian genes in stem cells? For instance in the subsection “Unexpected abundance of neoblast-specific retained introns”, the authors put forward a very interesting mechanism that is, nonetheless, speculative. The authors could word this more conservatively by mentioning ES data, then state these genes are good targets to test in planarian neoblasts to test for mechanistic similarity. Overall, given the embryonic strategy used by mammalian embryos to derive ES cells from embryos undergoing implantation, and that such an attribute is unlikely associated with the common ancestor of planarians and mammals, the homology between ES cells and neoblasts is dubious. Nonetheless, opposing roles for CELF/MBNL factors in stem/progenitor/embryonic cells, or cells associated with multi/pluripotency in their last common ancestor is certainly possible and would be very interesting. Such a role could have been variously recruited to embryonic cells and different stem cell classes in the course of evolution. This could be assessed by studying many stem cell/embryonic cell classes in other organisms/animal tissues. It is also possible that opposing roles in AS programs for CELF/MBNL proteins is what is conserved and that this could be variously utilized in different types of biology. We recommend a more careful/nuanced wording throughout, but particularly in the discussion. This may help bring the evolutionary line of argumentation into sharper focus.

We have now performed an evolutionary comparison between the MBNL targets in humans (described in Han et al., Nature 2013) and those we identified by knocking down the *mbnl* genes in planarian Xins fractions. We have found no conservation of specific exon targets, but we identified several genes with MBNL-regulated exons in both lineages (most of these overlap with the ES/neoblast-regulated exons, described in [Supplementary-material SD8-data]). It should be noted, however, that while we see a very good level of conservation in the wild-type samples from two planarian species (*Schmidtea mediterranea* and *Dugesia japonica)* conservation over such large evolutionary distances (i.e. planarians to mammals) is much harder to assess. Based on these results and following the Editor's suggestion, we have used more conservative phrasing when referring to the conservation of Mbnl regulation. Furthermore, we emphasize that the similarity is at the level of the regulatory axis (upstream factors) primarily and not the targets. This is for instance the case for miRNAs and their targets: while the upstream factors (miRNAs) are very well conserved there is extensive rewiring of the miRNA-target relationships. To reinforce this view, we have further elaborated the discussion by mentioning recent transcriptomic studies in animal stem cells and embryo development where CELF/MBNL antagonism could be at work in pluripotent cells, and more carefully worded the conservation case throughout. We believe that under the light of the recent findings of Alie and co-workers (PNAS 2015) – where they find expression of a CELF homolog in sponge (poriferan) stem cells (archeocytes) and an MBNL homolog expressed in all other differentiated cell types profiled – the case for conservation of CELF/MBNL antagonism in stem cells of all animals is stronger. However, this and other possibilities are now properly discussed in the text.

3) The model put forward in Figure 8 overinterprets the data. We find no clear evidence in the manuscript for "impaired differentiation". Additionally, the significance of the connecting lines and arrows is far from clear. Do these arrows represent phenotypes caused by RNAi loss of gene function or do they represent gene function? In other words, in Figure 8, bottom image, the arrow seems to imply that bruli functions to negatively regulate stem cell renewal, etc., but the proposed roles for bruli and mbnl are opposite to arrows. If the negative arrows are meant to reflect phenotype, that is an atypical and confusing way to summarize gene function. mbnl arrows reflect more interpretation than is possible from the data-the authors have not shown whether there is more progression from SC to prog cells or less progression from prog cells to differentiated cells.

We have addressed these issues in the new model presented; we have more carefully listed the observed phenotypic defects and eliminated the arrows.

4) The high conservation of tissue-specific splicing regulators and their regulatory codes has also been documented before (PMID: 21389270, 25576366, 20921232). The human CELF4 can rescue the function of C. elegans orthologue (PMID: 12906792) and the MBNL sequence specificity, its regulatory program and many of its binding sites are conserved between Drosophila and human (PMID: 22901804). The authors should describe this further in the manuscript.

We have introduced a new paragraph in the Discussion that highlights previous reports on the conservation of the binding affinities and regulatory codes of RBPs.

*5) Please confirm that all large scale transcription datasets are deposited in a publicly accessible repository.*

Confirmed.